## Registered report

psychology

prediction, error-based learning, language acquisition

**Author for correspondence:**
Judit Fazekas
e-mail: Judit.Fazekas@mpi.nl

# Do children learn from their prediction mistakes? A registered report evaluating error-based theories of language acquisition

Judit Fazekas[1], Andrew Jessop[1], Julian Pine[2] and Caroline Rowland[1,2]

[1]Language Development Department, Max Planck Institute for Psycholinguistics, Nijmegen, Gelderland, The Netherlands
[2]Psychological Sciences, University of Liverpool Institute of Psychology Health and Society, Liverpool, UK

 JF, 0000-0002-6059-4109

Error-based theories of language acquisition suggest that children, like adults, continuously make and evaluate predictions in order to reach an adult-like state of language use. However, while these theories have become extremely influential, their central claim—that unpredictable input leads to higher rates of lasting change in linguistic representations—has scarcely been tested. We designed a prime surprisal-based intervention study to assess this claim. As predicted, both 5- to 6-year-old children ($n = 72$) and adults ($n = 72$) showed a pre- to post-test shift towards producing the dative syntactic structure they were exposed to in surprising sentences. The effect was significant in both age groups together, and in the child group separately when participants with ceiling performance in the pre-test were excluded. Secondary predictions were not upheld: we found no verb-based learning effects and there was only reliable evidence for immediate prime surprisal effects in the adult, but not in the child group. To our knowledge, this is the first published study demonstrating enhanced learning rates for the same syntactic structure when it appeared in surprising as opposed to predictable contexts, thus providing crucial support for error-based theories of language acquisition.

# 1. Introduction

Prediction, the ability to anticipate other people's upcoming words or actions, plays a key role in a wide range of different human behaviours and activities, from making music [1] to playing volleyball [2]. Prediction plays such a central role in some theories of cognition that human brains have been described as 'prediction machines' [3, p. 81]. Prediction is particularly important in human communication. It has been suggested, for instance, that prediction can contribute to smooth turn-taking in conversation, not just because it enables us to anticipate when our partner will stop speaking and we can begin speaking ourselves, but also because, by successfully predicting upcoming words, we can give ourselves time to prepare an appropriate response [4]. Although some scholars question how central prediction's role in human communication really is [5,6], other theories go even further and claim that prediction is a key mechanism in language processing itself [7,8].

While the role of prediction in adult language use is well documented, there is also the further possibility that prediction is not just vital for using language, but also for acquiring it in childhood. This is the basis of error-based theories of language acquisition. Error-based theories (which can explain learning patterns outside of language as well [9,10]) suggest that children, like adults, continuously predict upcoming words in conversation, and use these predictions to build up their competence in their first language by comparing what they predicted with the actual input received [11,12]. One such model, the frequency-based, connectionist Dual-path model [11], uses an error-based learning mechanism [13] to model the acquisition of syntax, the developmental phenomenon that is the focus of the current study. In this model, if there is a discrepancy between the predicted and actual syntactic structure, an error signal is generated, which is then used to adjust the weights that support syntactic knowledge. These weight changes accumulate over time and allow children's syntactic knowledge to gradually approximate the adult state (note, however, that this is not a stage-based theory; the process also results in representational change in adults, but less obvious change because adults' representations are less malleable).

There are several reasons why error-based theories of language acquisition have gained wide support. First, they provide an interactive model that treats language acquisition as the outcome of processing. According to error-based theories, children (and, in fact, adults) constantly predict words and evaluate predictions while processing language. Every time they make an incorrect prediction, linguistic representations change, which, in children, moves them a step closer to the adult state. This means that error-based theories allow for the possibility that limitations in processing might influence acquisition. Second, rather than simply seeking to define children's state of knowledge at different developmental stages, these models explain how children move from one knowledge state to another. For instance, the Dual-path model [11] not only describes the error-based learning mechanism (that adjusts weights supporting linguistic knowledge in response to error signals), but also demonstrates how this mechanism leads to changes in performance over development (from being able to identify agent and patient roles in intermodal preferential looking experiments at an early age, to producing correct sentences with novel verbs later on). Error-based learning theories thus provide a specific learning mechanism that can be tested experimentally. Third, models implementing error-based learning mechanisms are supported by experimental data and provide explanations for developmental phenomena that are challenging for earlier language acquisition theories. For instance, an error-based noun-acquisition model proposed by Ramscar *et al.* [12] explains how overgeneralized forms (like 'mouses') disappear from children's productions in the absence of explicit correction. When children predict the overgeneralized 'mouses' form but hear 'mice' instead, the associations between the plural of 'mouse' and 'mouses' weaken due to the error signal resulting from the incorrect prediction, while associations with 'mice' are strengthened. Over time, children start producing and predicting the more strongly associated 'mice' form instead of 'mouses'.

Despite widespread enthusiasm for theories that embrace the role of prediction as a learning mechanism, there remains a major problem. There is to date only limited evidence that children actually do generate linguistic predictions, and what evidence there is does not show that these predictions are used for learning. The most promising aspect of error-based theories—that they propose a viable and intuitive language learning mechanism—has therefore yet to be systematically tested. The goal of the present study is to examine the role of prediction in language acquisition by assessing whether less predictable (more surprising) input leads to more lasting change than more predictable input. Below, we review the current state of the literature, particularly previous developmental studies on prediction, before discussing the aims of the current study in more detail.

Language acquisition plays a central role in developmental research on prediction, and several experimental studies assess the relationship between prediction and learning. Some studies

concentrate on the relationship between predictive abilities and certain aspects of language proficiency [14–17]. For example, Mani & Huettig [14] found that toddlers' prediction skills (measured using a version of the preferential looking paradigm) significantly correlated with their productive, but not their receptive vocabulary. Other studies have assessed the nature of children's linguistic predictions in order to examine whether they could form the basis of learning [18–21]. Gambi *et al*. [19] found that children can use semantic associations as a basis for their predictions [18] and combine them with predictions based on syntactic knowledge [22], showing that children's predictions could be a viable basis for language acquisition.

Studies targeting prediction in childhood typically use the visual word paradigm and have been successful in demonstrating that children use anticipatory eye gazes to visual scenes to predict upcoming words in sentences. However, they do not investigate whether this effect then leads to subsequent learning. They only study whether or not children make predictions; they do not examine whether the learning mechanism compares these predictions with actual input or whether the outcome of this comparison leads to subsequent language change. In other words, this paradigm does not address whether predictions form part of an error-based learning mechanism.

There is also another, perhaps more fundamental, problem with using the visual world paradigm to study prediction. This is that so-called predictive looking could, in fact, be a result of a process of integration. In these studies, children listen to sentences where the final word is highly predictable, while their eye movements on an array of pictures are recorded [14,18,19]. Such studies have shown that children as young as 2 years old tend to look longer at pictures of objects that would be a more predictable ending for the sentence after hearing the verb, but before hearing the last word [14]. For example, they are more likely to look at a picture of a cake rather than a picture of a stone after hearing 'The boy eats the big …', that is, before the sentence has been completed. These looks are referred to as anticipatory gazes and are regarded as evidence for prediction. However, according to Rabagliati *et al*. [23], it is possible that these effects are the result of integration and not prediction. If so, children would be looking at the picture of a cake after hearing *eat* because they chose *cake* as the most fitting sentence ending among the given picture alternatives, not because they predicted *cake* themselves. This means that instead of pre-activating upcoming words, children simply incorporate words based on the available visual input (see a similar discussion in the context of EEG research [24]). If so, these studies might not be providing an accurate measure of children's predictions.

In summary, while some studies have shown a correlation between prediction and learning, and others have shown the potential for prediction to act as a learning mechanism, no studies, to our knowledge, have directly assessed whether predictions lead to lasting changes in underlying linguistic representations—that is, whether they actually contribute to learning in children (though see [25,26] for adult participants). In addition, doubts have been expressed in the literature about whether the visual word paradigm really measures prediction or integration.

Our study aims to directly investigate both of these issues. We tested whether predictions lead to language learning in childhood using a novel method—prime surprisal [26,27]—to assess whether less predictable linguistic input leads to more lasting language change than more predictable input.[1] This method not only provides us with information about the immediate and longer term outcome of correct and incorrect predictions, but also overcomes the problems inherent in using the looking-while-listening paradigm, as it does not involve pictures of more or less predictable sentence endings, and so the responses cannot be guided by visual input.[2]

Prime surprisal studies are based on the priming paradigm [28,29], which is often used to examine syntactic development [30,31]. In priming studies, participants are exposed to a prime sentence involving a particular syntactic structure (e.g. active or passive), and then asked to respond to a target stimulus (e.g. a video that they must describe). If participants re-use the previously processed structure, especially if prime and target sentence share no content, this shows that they have access to the shared (abstract) structural representation underlying the prime and target sentence. This

---

[1]Footnote added after Stage 2 review: Within the Dual-path model (that is tested here), any non-immediate priming effect (even one that lasts for only a few intervening sentences) must reflect long-term weight changes (i.e. learning), since immediate activation effects decay instantly. In other words, learning is defined as weight change in the model and delayed priming is a manifestation of that weight change. The goal of the current study was to contrast immediate priming effects (measured directly after the prime sentence with minimal or no intervening linguistic input between prime and target) with lasting effects (that persist over multiple intervening sentences involving stimuli that contain the structure that has been primed, here datives).

[2]Footnote added after Stage 2 review: While the current study also used visual input (videos that the participants describe), we have not used this to measure predictive processes. The videos were only included to encourage participants to produce comparable dative sentences.

methodology has been particularly useful in demonstrating at what age children develop knowledge of different, abstract syntactic structures. Prime surprisal takes this method a step further by contrasting priming effects in response to predictable and surprising stimuli.

Prime surprisal studies typically feature syntactic structures that can appear in different forms with similar meanings. Dative structures, for instance, appear both as prepositional datives (PD, e.g. 'The student gave the report to the teacher') and double object datives (DOD, e.g. 'The student gave the teacher the report'). While DODs appear more often in adult language use overall, every verb has its own specific preferences: for instance, while the verb *give* occurs more often in a DOD structure than in a PD structure, the verb *bring* prefers the PD structure. Children need to acquire these links in order to produce well-formed sentences and avoid incorrect verb-structure pairings (such as '*the student spoke her teacher the answer').

Prime surprisal studies with both child and adult participants have found enhanced priming effects when a structure appeared with a mismatching as opposed to a matching verb [26,27]. According to the Dual-path model, these effects result from the error-based prediction mechanism: after hearing a verb, children predict the dative structure that most often follows that particular verb. If they end up hearing a different structure to the one they predicted, the learning mechanism produces an error signal, which they then use to adapt their syntactic knowledge accordingly. In a previous prime surprisal study [27], for instance, priming effects were larger when a DOD structure appeared with the verb *bring* (PD-biased) than when it appeared with the verb *give* (DOD-biased), without verb repetition between prime and target sentences. According to the Dual-path model, this occurs because, in the mismatching condition (e.g. DOD with *bring*), participants are likely to make a prediction error. They are likely to predict that the PD-biased verb will be followed by the structure that appears more often with that verb (PD). For example, after hearing 'the boy brings…' participants are more likely to predict '…*the present to the girl*' (PD) than '…*the girl the present*' (DOD). Since this prediction will turn out to be incorrect, an error signal will be generated, which will, in turn, lead to a change in the weights supporting syntax and to a higher likelihood of the participant reproducing the structure that they have just heard. No such effect occurs in the matching condition: here, when a structure appears with a matching verb (e.g. DOD with *give*), the participants are more likely to successfully predict the upcoming structure, which means that no error signal will be produced. In other words, according to the Dual-path model, the error signals and weight changes that lead to immediate prime surprisal effects are actually a consequence of the long-term learning that will eventually result in adult-like syntactic preferences.

Although the verb-structure links leading to prime surprisal effects form a key part of syntactic knowledge, they are not fully adult-like at 5–6 years of age. According to error-based learning theories, children make predictions from early on, but these early predictions are based on limited linguistic input and therefore are more often incorrect. The older children are, the more adult-like their language becomes and the more correct predictions they make. At the age of five, children have already accumulated enough knowledge to have verb-structure preferences similar to those of adults, but since these preferences are based on less linguistic input, they are weaker and more malleable. Children's weaker representations lead to stronger priming effects [27] and, according to error-based theories, more learning as well. By contrast, the more developed adult system is less sensitive to the error signals produced by unexpected sentences, resulting in smaller priming and learning effects.

Prime surprisal effects provide promising evidence for prediction in both children and adults, and suggest that incorrect predictions influence subsequent behaviour in the short term. However, the key prediction of this account is that incorrect predictions lead to learning. To test this, we need to demonstrate that prime surprisal leads to lasting cumulative language change as well. To do this, we have developed a new design which combines the prime surprisal method with a paradigm designed to assess whether the original priming effects are cumulative and persistent (see Kaschak *et al.*'s work [32] for an adult study). Studies in this paradigm typically start with a baseline phase where participants' unbiased rates of the target construction are assessed (e.g. how many DODs and PDs they produce), followed by a test or bias phase where participants are biased towards the production of one of the structures (e.g. are only exposed to PDs or DODs). Finally, in a post-test phase, participants' rates of target construction are re-assessed to see whether they have shifted towards the structure they were biased towards in the previous phase.

Developmental studies using similar designs have shown that children's production frequencies can be shifted towards a less frequent structure by exposure in the bias phase [33–35]. These results are in line with the predictions of the Dual-path model, but, due to the set-up of these experiments, they could have originated from sources other than error-based learning. For instance, some studies did not contrast the

effects resulting from experience with less-expected structures with the effects resulting from experience with more-expected structures, in which case the post-test shift could be the result of cumulative facilitation from processing a structure multiple times rather than error-based learning [34,35]. Other studies included primes in the post-test phase (as well as the bias phase), meaning that the effects from the bias phase and those of immediate priming are measured on the same target items, making it difficult to tease apart long- and short-term effects [33]. The implication is that the strong prediction of the Dual-path model that less predictable (i.e. more surprising) linguistic input leads to more lasting language change still needs to be systematically tested.

We conducted a four-phase experiment with child and adult participants featuring both predictable and surprising structures in the bias phase, and only including target structures in the baseline and post-test phase. This way, we were able to directly contrast lasting language change resulting from more- or less-expected structures, and clearly differentiate between immediate and lasting effects of predictability. Furthermore, instead of simply contrasting effects of overall more- or less-expected structures (e.g. DODs versus PDs), we contrasted the effects of the same structure presented in a more or less predictable environment (by consistently presenting PD and DOD structures with either matching or mismatching verbs in a within-participant design, table 1).

This allowed us to get clearer results from the child participants, whose overall dative preferences are inconsistent and not yet adult-like [36], but who have already been shown to be sensitive to verb-bias effects [27]. Furthermore, by featuring the same number of PD and DOD structures in both conditions, and only varying how likely it is that participants correctly predict them, we could ensure that the potential differences between results in each condition are due to differences in predictability.

In sum, error-based models that posit prediction as a learning mechanism provide a very promising avenue for understanding the language acquisition process. However, there is limited evidence for the existence of linguistic prediction in childhood, and its contribution to learning has not been systematically examined. To our knowledge, this is the first study that directly targets the role of prediction in language development by assessing whether unpredictable input leads to more lasting language change than predictable input.

## 2. Methods

The goal of this study was to examine the role of error-based learning in acquisition by assessing whether less predictable (more surprising) linguistic input leads to more lasting language change than more predictable input. To achieve this, we used the prime surprisal paradigm in a four-phase experiment, designed to induce error-based learning via prime surprisal. The prime surprisal paradigm capitalizes on the fact that some verbs are substantially more likely to appear in one dative sentence structure than another in English and are thus surprising, despite being grammatical, in the alternative structure. Error-based learning predicts a bigger change in children's syntactic representations (i.e. learning) after surprising (e.g. PD-biased verb in a DOD structure) than unsurprising (DOD-biased verb in a DOD structure) primes.

Learning is defined as a change in the underlying syntactic representations and is operationalized as a change in performance from pre- to post-intervention in a production task. More specifically, learning was deemed to have occurred if the children were significantly more likely to use the primed dative structure post-intervention than pre-intervention (i.e. there was a change in the strength of the children's underlying syntactic representations induced by the priming).

In the first, baseline phase of the study, we assessed participants' baseline rates of dative production (i.e. how many DODs and PDs they produced). Participants described target video animations depicting transitive actions with dative sentences, but were free to choose either PD or DOD structures, and the experimenter described filler videos depicting non-causal actions that could be described with intransitive sentences.

The second, priming (or bias) phase was designed to elicit immediate prime surprisal effects [27] and biased the participants towards one of the dative structures. Here, participants described target video animations depicting transitive actions in a similar way to the baseline phase, but the experimenter preceded these participant descriptions by describing prime animations using either DOD or PD structure. Both structures were consistently paired with either matching or mismatching verbs in the prime sentences (e.g. PDs only appeared with matching verbs, while DODs only appeared with mismatching verbs for group A and vice versa for group B). This way, participants in group A were always subjected to PDs in predictable sentences and DODs in surprising sentences.

**Table 1.** General study design showing different trials and verb biases in each phase—in the bias phase dark grey cells signal surprising prime sentences while light grey cells stand for predictable primes. When the structure is specified as 'DOD' or 'PD', the experimenter produces a full (DOD or PD) dative, and when it's specified as 'dative' the participant completes a sentence stem with their choice of a dative structure.

| | group A DOD-bias | | group B PD-bias | |
| --- | --- | --- | --- | --- |
| | **structure** | **verb bias** | **structure** | **verb bias** |
| *baseline phase* | | | | |
| **experimenter** | filler | NA | filler | NA |
| **participant** | **dative** | **equi-biased** | **dative** | **equi-biased** |
| **experimenter** | filler | NA | filler | NA |
| **participant** | **dative** | **equi-biased** | **dative** | **equi-biased** |
| *bias phase* | | | | |
| experimenter | DOD | PD-biased | DOD | DOD-biased |
| **participant** | **dative** | **PD-biased** | **dative** | **PD-biased** |
| experimenter | PD | PD-biased | PD | DOD-biased |
| **participant** | **dative** | **DOD-biased** | **dative** | **DOD-biased** |
| experimenter | DOD | PD-biased | DOD | DOD-biased |
| **participant** | **dative** | **PD-biased** | **dative** | **PD-biased** |
| experimenter | PD | PD-biased | PD | DOD-biased |
| **participant** | **dative** | **DOD-biased** | **dative** | **DOD-biased** |
| *post-test phase* | | | | |
| **experimenter** | filler | NA | filler | NA |
| **participant** | **dative** | **equi-biased** | **dative** | **equi-biased** |
| **experimenter** | filler | NA | filler | NA |
| **participant** | **dative** | **equi-biased** | **dative** | **equi-biased** |
| *second post-test phase* | | | | |
| **experimenter** | filler | NA | filler | NA |
| **participant** | **dative** | **PD-biased** | **dative** | **DOD-biased** |
| **experimenter** | filler | NA | filler | NA |
| **participant** | **dative** | **PD -biased** | **dative** | **DOD-biased** |
| **experimenter** | filler | NA | filler | NA |
| **participant** | **dative** | **PD-biased** | **dative** | **DOD-biased** |
| **experimenter** | filler | NA | filler | NA |
| **participant** | **dative** | **PD-biased** | **dative** | **DOD-biased** |

The third, post-test phase was similar to the baseline phase, but the goal was to reassess participants' rates of dative production. If less predictable input leads to more lasting language change than more predictable input (as suggested by error-based learning theories), we expected participants' production in this phase to shift towards the structure they were exposed to with a mismatching verb in the bias phase (i.e. DODs for participants in group A) compared with the baseline phase. In order to eliminate the influence of lexically based long-term priming effects, we used different verbs in the bias and test phases.

While the main focus of this study was abstract error-based learning, the second post-test aimed to assess potential verb-specific learning effects. This phase was similar to the pre- and post-test phases, but the target sentences uttered by the participants re-used the PD- or DOD-biased verbs that were featured as primes in the bias phase. This way, we were able to detect a possible change in participants' verb-specific syntactic representations without interfering with the abstract priming

effects in the previous phases. If there is verb-specific error-based learning, we expect an enhanced shift towards the dative structure the verb previously appeared with when the structure did not match the verb's bias. For instance, for the PD-biased verb *bring*, we expected a bigger shift towards the structure for participants for whom it consistently appeared with the mismatching DOD structure than for participants for whom it appeared with the matching PD structure.

Unpublished results from Fisher & Lin [37] show that training with less-expected sentences can lead to larger shifts in dative production than training with more predictable sentences if the verb is shared between training and test. Replicating these results in our study would serve as a good basis for comparison with our main focus, abstract error-based learning. The current study was pre-registered on the Open Science Framework (OSF); the accepted Stage 1 registration can be viewed at (https://osf.io/khym8/).

## 2.1. Participants

Seventy-two 5- to 6-year-old children (47 female, mean age 76.15 months, s.d. = 9.59 months) and 72 adults (62 female), all monolingual English-speakers, participated in the study. The child participants were recruited from schools in the area and the departmental database, while the adult participants were recruited from the university's student participation pool.

Ten child and two adult participants who produced 'other' responses for more than half of the target trials in the test, post-test or second post-test phases were excluded. These participants were replaced in order to obtain 72 sets of data in each age group. Exclusion criteria for the target sentences will be discussed in the §2.7 Coding.

These age groups have shown sensitivity to verb-bias manipulations both in the target verb and in the prime verb (prime surprisal) conditions in a priming study involving dative structures [27]. Children of this age consistently produce both PD and DOD structures (with an average DOD production of approximately 30%) in corpus-based studies [38], and similar frequencies were observed in priming studies using a similar paradigm to our own [27,31]; therefore no floor or ceiling effects were expected to occur in this study.

We determined our sample sizes based on power calculations carried out to allow both of our key comparisons of interest and our manipulation check to be powered adequately. We carried out two sets of power calculations across 1000 iterations on simulated binomial data using mixed effects models, based on those that were used to carry out analyses on our observed data (see §3 Statistics and data analyses). Maximal models were fitted to the simulated data. If the model failed to converge on 20% of the simulations, it was rejected and simplified before the power analysis was repeated. As our main point of interest in this study was the performance of the child participants, our calculations were based on the effect sizes expected in this group.

Our first power calculation was carried out on our key comparison of interest assessing whether less predictable (more surprising) input leads to more lasting syntactic representation change than predictable input (see power calculation: https://osf.io/9ecjh/ and details of the analyses this calculation is targeting in §3.1). As there are currently no data available for our main comparison in the literature, we estimated our simulated effect sizes based on studies targeting contrasts that are in some respects similar to ours, such as 4-year-olds' post-intervention performance in a study involving the passive structure [34], an adult intervention study featuring the dative structure [32] and a developmental study involving 5- to 6-year-olds looking at immediate prime surprisal effects featuring the dative structure [27]. The effect sizes most relevant to our comparison in the following studies were 11% post-test shift in a passive intervention study with 4-year-olds [34], an average 7% post-intervention shift in a dative study featuring adults [32] and 16% higher priming after mismatching primes than matching ones in an immediate prime surprisal study in 5- to 6-year-olds [27]. Based on the above results, we expect at least a 10% shift in both bias groups towards the structure participants were biased towards in the bias phase. In order to ensure that the study was adequately powered even if there were smaller than expected effect sizes, we estimated an average 5% shift in both bias groups (showing that participants' production in the post-test phase shifts towards the structure they were exposed to with a mismatching verb in the bias phase). Based on corpus-based studies [38] and priming studies using similar materials to our own [27,31], we estimated an average 30% baseline DOD production in the pre-test phase in both bias groups. Our power calculation showed that our key comparison of interest (post-test differences based on bias group captured by the prime-bias variable) had 93% power when featuring 66 participants. We planned to

include 72 participants in each age group in order to have equal numbers of participants in the eight counterbalance groups and to account for 10% potential data loss.

We also carried out a separate power calculation to ensure that our manipulation check (immediate prime surprisal effect in the test or bias phase, see power calculation: https://osf.io/x2ykf/ and details of the analyses this calculation is targeting in §3.2) was adequately powered. As this phase aimed to replicate the effects in Peter *et al.*'s study [27], we simulated data based on the response frequencies in the 5- to 6-year-old group. We estimated an average DOD production of 24% and 35% in the matching PD and DOD prime conditions and 19% and 41% in the mismatching PD and DOD prime conditions. Our power analysis targeted the interaction of prime structure and verb bias. Based on these estimates, the power analysis returned 81.3% power when including 66 participants. With the inclusion of an extra six participants (to account for 10% potential data loss), this phase of the study was therefore also sufficiently powered.

## 2.2. Design

The between-subject variables were age (adults versus children) and bias group (DOD-bias and PD-bias), and the within-subject variables were verb-bias match (match or mismatch), prime type (DOD and PD) and phase (pre-test, bias phase, post-test and second post-test). The dependent variable was the choice of dative structure in the target trials.

## 2.3. Predictions

We had four main predictions, which are discussed in more detail in §3 (Statistics and data analyses).

1. **Immediate prime surprisal:** we expected to replicate the effects found in Peter *et al.*'s study [27] and find increased priming if the verb bias and the prime structure did not match in the prime sentence.
2. **Learning about abstract structures:** we expected that less predictable (more surprising) input would lead to more learning than predictable input. Therefore, we expected that participants' production in the post-test would shift towards the structure they were exposed to with mismatching verbs in the bias phase.
3. **Verb-based learning:** due to the larger learning effects resulting from unpredicted input, we expected that participants would be more likely to re-use the structure the target verb previously appeared in if that structure did not match the verb's bias.
4. **Stronger effects in the child than in the adult group:** due to the weaker and more malleable verb-biases in children compared with adults, we expected that the three above effects (immediate prime surprisal, learning about abstract structures and verb-based learning) would be larger for children than adults.

## 2.4. Visual stimuli

The study featured video animations created in Moho 12, which were presented in E-prime 2.0 software [39]. Each participant saw 120 videos: 60 videos depicting transitive actions that can be described with prepositional or double object datives for the prime and target sentences and 60 videos depicting non-causal actions for the filler sentences.

The cartoons included 10 pairs of donor and recipient characters. Half of them were cartoon characters that are familiar to British children with proper noun names: *Tigger and Piglet*, *Dora (the Explorer) and Boots*, *Marge and Homer*, *Lisa and Bart* and *Bob (the Builder) and Wendy*. The other characters were referred to with determiner and noun NPs: *the prince and the princess*, *the king and the queen*, *the student and the teacher*, *the doctor and the nurse* and *the boy and the girl*. Particular donor and recipient characters were always featured together. A further 10 items acted as objects and were referred to with indefinite determiner and noun NPs: *a ball*, *a toy*, *an orange*, *a cake*, *a peach*, *a sandwich*, *a pencil*, *a book*, *a napkin* and *an apple*. The objects were consistently paired with one pair of characters (e.g. the *ball* was always featured with *Bob and Wendy*).

In the bias phase, prime videos were always paired with a target video that included different characters from those in the prime. In order to control for the possibility that direction of transfer might influence structure choice, the animations depicted the direction of motion of transfer actions equally often from right-to-left and from left-to-right.

## 2.5. Sentence stimuli

The study contained 120 sentences (including 60 verb stems) per participant: 16 prime and 16 target sentences plus 32 fillers in the bias phase, 10 target and 10 filler sentences in the pre- and post-test phases and 8 target and 8 filler sentences in the second post-test. The prime sentences appeared half the time as DOD sentences and half the time as PD sentences. Both structures were consistently paired with either matching or mismatching verbs in the prime sentences (e.g. PDs only appeared with matching verbs while DODs only appeared with mismatching verbs for participant A and vice versa for participant B). The target sentences were produced by the participant (as either DOD or PD sentences) based on the video stimuli.

For instance, a prime-target trial in the bias phase included a prime sentence such as 'The king brought the queen a cat.' (DOD) or 'The king brought the cat to the queen.' (PD) and participants completed a sentence stem such as 'Lisa dropped…' as a target sentence.

In order to avoid lexically based long-term priming effects, we used a different set of verbs in the bias phase and in the pre- and post-test phases. The study involved the following two sets of verbs, featured here with their DOD frequencies in the Manchester corpus [40] in brackets (for the computation of the dative frequencies see [41]). The first set of verbs was used in the pre- and post-test phases. This set contained three equi-balanced verbs: *feed* (52%), *slide* (56%) and *throw* (49%), and one PD- and one DOD-biased verb: *bring* (27%) and *give* (89%). The second set of verbs was featured in the test phase and repeated in the second post-test. This set contained four PD-biased verbs: *leave* (32%), *sell* (24%), *send* (44%) and *take* (15%) and four DOD-biased verbs: *award* (83%), *hand* (63%), *offer* (77%) and *show* (93%).

We selected the above verbs based on the frequency of their dative occurrences in the Manchester corpus [40]. These verbs have yielded immediate prime surprisal effects in other studies featuring similar age groups to ours [27] as well as in our pilot study featuring 5- to 6-year-old children.

We aimed to select verbs that had strong verb biases for the bias phase (as prime surprisal is defined as the negative logarithm of the verb bias [25]), but our choices were constrained by the limited number of verbs that appeared often in dative structures in the Manchester corpus [40].

To control for sentence-specific preferences, we created eight counterbalance groups to ensure that (i) if the DOD structure consistently appeared with matching verbs in one counterbalance group, it appeared with mismatching verbs in the other (and vice versa for the PD structure), (ii) if a verb appeared with a DOD in a counterbalance group, it appeared with a PD in the other, and (iii) if a target sentence appeared in the pre-test in one counterbalance group, it appeared in the post-test in the other.

Semi-randomized[3] stimulus lists were created in which the prime and target sentences always followed each other in the bias phase and the same verb did not appear twice in immediate succession. In the test or bias phase, there was always a pair of filler sentences after every target sentence. In the other phases, filler and target phrases alternated with each other.

## 2.6. Procedure

The study used the bingo game paradigm [27,31]. It took the form of a bingo game in which experimenter and child took turns to describe cartoon animations or pictures on a laptop computer.

The experimenter introduced the characters involved in the tasks by showing the participants cards featuring the characters. The experimenter and the participant sat in front of the computer side by side.[4] The experimenter described the first cartoon and asked the participant to repeat the sentence. The participant was then asked to produce a target sentence by describing a cartoon animation on the other side of the screen. To ensure that participants' responses contain the target verb, a stem-completion technique was used (e.g. *the boy gave…*) [27].[5] Each target sentence was immediately followed by an intransitive filler sentence.

---

[3]Footnote added after Stage 2 review: We used a semi-randomized approach to ensure that lists adhere to the above criteria.

[4]Footnote added after Stage 2 review: Before the study began, the participants were given the following instructions: 'We will be watching videos and describing them to each other. When the video appears on my side of the screen (experimenter points to the left side of the screen) I will be describing the video to you and you will have to repeat what I said. When the video appears on your side (experimenter points to the right side of the screen) you will describe the video to me, but I will always start the sentence for you. You will have to repeat what I said, and finish the sentence. Sometimes we will see a happy or a sad face. If it's a happy face, we get to pick a card and check whose Bingo board it belongs to. If it's a sad face, we don't pick a card that time.'

[5]Footnote added after Stage 2 review: The experimenter described the videos instead of using pre-recorded materials as the Bingo paradigm relies on the interaction between participant and experimenter to keep the child engaged through the study.

After every two or three sentences, a smiley or frowny face appeared to signal whether a bingo card was available. If it was, the child or the experimenter got the card and could add it to their bingo grid. The first person to fill the bingo grid with bingo cards was the winner of the game, and the experiment was designed so that the participant always won.

Before beginning the study, there was a practice session to ensure that the participants understood the task. The practice session included intransitive sentences featuring three characters each (e.g. 'The king and the queen were playing with the cat.'). In order to encourage the production of full datives in the main study, we asked participants to mention all three characters in their descriptions during the practice session. To further encourage the production of full datives in the study, the first verb featured as a target in both the pre- and post-test phase was a verb that cannot be used as an intransitive.

The bingo paradigm paired with the stem-completion technique has been successfully used to elicit dative sentences in similar age groups and has resulted in low exclusion rates [27,31]. Furthermore, both the child and adult participants enjoyed participating in our pilot study featuring this paradigm and all participants completed the session.

After completing the bingo game, we measured children's baseline language abilities following a Stage 1 reviewer's request. As we aimed to capture individual differences in children's morphosyntactic abilities, we initially planned to use the Sentence Imitation Task from the Early Repetition Battery [42] (SIT). However, as members of our research group have found ceiling effects with a similar population to the one included in our study using SIT, we proposed using the Test for Reception of Grammar 2 [43] (TROG) instead. After discussion with the editor, we administered both tests, but as a ceiling effect— defined as over 70% of the children providing a correct answer for at least 25 out of the 27 items— occurred in the SIT, we included only the children's TROG scores in our analyses. The study lasted approximately 45 min, including a break, and participants received a sticker after the practice session.

## 2.7. Coding

The experiment was audiotaped, allowing the transcription and coding of the utterances off-line. The first author transcribed the utterances. Then, two coders who were both blind to the experimental condition coded them. The first coder coded all utterances and a second coder coded 10% of the utterances in order to compute the Cohen's kappa inter-rater reliability [44]. Inter-rater reliability was high at 99.8% agreement, Cohen's kappa = 0.99. Coders resolved potential discrepancies by revisiting the sentences in question and the mutually decided code was included in the dataset.

A target response was considered a DOD if it contained the correct target verb followed by two noun phrases, and a PD if it contained the correct target verb followed by a noun phrase and a prepositional phrase headed by 'to'. Responses were coded as 'other' if (i) the participant failed to repeat the prime correctly (even after help), (ii) if the participant produced a non-target verb, or (iii) if the target sentence could not be classified as a DOD or PD response based on the above criteria (e.g. target responses containing a preposition other than 'to' or incomplete datives such as *'the king gave the ball'*).

# 3. Statistics and data analyses

The data were analysed in R version 3.6.3 [45], through a series of logistic mixed effects models [46,47] fit using the lme4 1.1–23 package with the *nloptwrap* optimizer. These models were initially specified with subject and item as random grouping factors, each including all of the relevant within-subject and within-item fixed effects as random slopes with their associated correlation parameters. Where necessary, these models were simplified until there were no issues with convergence or singular variance-covariance matrices. The models were then assessed for overparametrization using a principal components analysis on the random effects structure, with further simplifications being performed if required [48,49]. In all models, the dependent measure was the binomially coded production of DOD structures (DOD = 1, PD = 0). The factors were coded with effect/sum contrasts [50,51], while age (in months) was entered as a continuous variable and was centred to reduce multi-collinearity [52]. Confidence intervals (Est. [CI]) for the model estimates were obtained using parametric bootstrapping ($r = 1000$). The confirmatory tests of the hypotheses and their *p*-values were obtained by sequentially removing individual contrasts from the fixed effect structure and running log-likelihood-ratio tests ($\chi^2$). However, we did not remove any fixed effects for the purpose of model selection or criticism; all fixed parameters were retained, even when they did not improve the model's goodness of fit.

At the request of a Stage 1 reviewer, we also performed a parallel series of Bayesian mixed effects models to match the frequentist analyses. These were implemented using the *rstanarm* 2.19.3 package, which provides front-end functions for using Stan [53] in an R environment. As we did not describe the Bayesian analyses in detail in the registered report, these are regarded as exploratory analyses. Consistent with the frequentist analyses, we first attempted to include all relevant within-subject and within-item fixed effects as random slopes. The models were then simplified to address any issues with convergence or an excessive number of divergent transitions when the target average acceptance probability was set at 0.99. In some cases, it was necessary to remove control covariates (e.g. TROG score) from the fixed effect structure to reach a model specification supported by the data. However, we only considered removing fixed effect parameters when their variance estimates were close to zero and the random effects structure could not be simplified any further. None of the parameters that directly address our core hypotheses were removed. In accordance with the recommendations for binomial outcome measures [54], we used weakly informative priors on a Student's *t* distribution for the model intercept and predictors. The algorithm took 10 000 posterior estimates of each parameter (5000 samples across four Markov chains, with a warm-up of 2500 samples). We report the mean posterior Beta estimate, 95% credibility intervals (mean [95% CrI]), and the posterior probability of the parameter estimate being larger than (for positive estimates) or smaller than (for negative estimates) zero (*P*).

A key advantage of Bayesian analyses, compared with the frequentist approach, is that the interpretation of the Bayesian probability estimates (*P*) and credible intervals (CrI) is more intuitive than that of the frequentist *p*-value and confidence intervals [55–57]. Bayesian posterior probability allows us to determine the probability of the true effect being different than zero, given our data (without any reference to a null hypothesis), while credible intervals identify the upper and lower bounds of where the true mean lies with 95% certainty (for a 95% CI). Thus, Bayesian analyses allow us to make statements about the likelihood of an effect given the data, in a way that is not possible based on frequentist estimates. It is worth noting that while Bayesian and frequentist approaches allow us to quantify our effects in different ways, they tend to lead to similar conclusions when used with weak and uninformative priors (such as the ones used in our analyses).

We carried out three sets of analyses on different subsets of the response data to (i) explore whether less predictable (more surprising) linguistic input leads to more persistent language change than more predictable input with no repetition of verbs (our main hypothesis), (ii) assess whether we replicated the prime surprisal effects found in Peter *et al.* [27], and (iii) explore whether less predictable (more surprising) linguistic input leads to more lasting language change than more predictable input for repeated verbs. We assessed our fourth hypothesis (iv) that stronger effects would be observed in the child than in the adult group in each section separately to determine which learning or priming effects are different in the two age groups. In the following sections, we describe all analyses involving data from both age groups together, but in order to explore the group-specific patterns in more detail, we also carried out analyses on the data from the two age groups separately. The main effects of age in months and TROG score (centred and rescaled) were added to the models examining data from the child group separately. All analysis scripts and relevant datasets can be accessed on https://osf.io/r8exu/, and an overview of the analyses can be found in figure 1.

## 3.1. Key comparison of interest—abstract learning effects

**Hypothesis 1—H2a—participants' production in the post-test shifts towards the structure they were exposed to with mismatching verbs in the bias phase and H2b—the shift described in H2a is stronger in the child than in the adult group**

This analysis tested the central prediction of error-based learning theories: that less predictable (more surprising) input leads to higher rates of lasting syntactic representational change than predictable input by testing whether the post-test scores differ in the two bias groups, while controlling for the pre-test performance. (Note that this is the second prediction presented in §2.3. Above, but we present it first here as it was our key analysis.) It was carried out on the target items from the post-test phase and the full model included (i) bias group (depending on whether participants were biased towards DOD or PD structures in the bias phase), (ii) pre-test score (how many DODs per datives a participant produced in the pre-test phase), and (iii) age group (children or adults, in the combined model), as fixed effects, by-subject random intercept with no random slopes and by-item random intercept with random slopes for bias phase, pre-test score and age group, in the combined analyses. If participants are influenced by input predictability, we expected to find a main effect of bias group showing that participants' dative production in the post-test phase is different in the two bias conditions and that

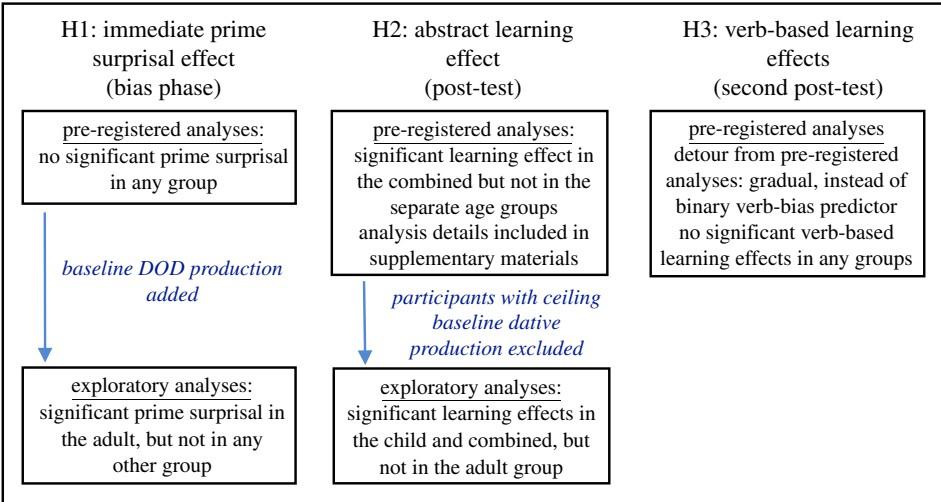

**Figure 1.** Analysis flowchart detailing pre-registered and exploratory analyses in the different stages of the study. All analyses were carried out using both frequentist and Bayesian mixed effects models.

they shifted towards the structure they heard with a mismatching verb in the bias phase (H2a). We also expected a stronger pre- to post-test shift in the child than in the adult group, demonstrated by an interaction between bias group and age group (H2b).

We carried out two sets of analyses here, our pre-registered analysis and an exploratory analysis, where we excluded participants who showed ceiling performance in the baseline phase. In the pre-registered analysis, we found that both age groups showed a pre- to post-test shift towards the dative structure they were exposed to in surprising (as opposed to predictable) sentences in the bias phase. This result is in line with the central claim of error-based theories: that unpredictable input leads to higher learning rates than predictable input. In the full dataset, this difference was significant in the frequentist analysis and the Bayesian posterior probability was high (96.88%), supporting Hypothesis 2a. However, the effect did not reach significance in the either of the age groups separately, though the Bayesian posterior probability was high, especially in the child group (adult group = 82.77%; child group = 91.22%). In addition, despite the numerically larger shift in the child compared with the adult group, the interaction of bias group and age group did not reach significance. Thus, we found no reliable support for Hypotheses 2b in our pre-registered analysis. As per a reviewer's request, the pre-registered analysis is only discussed in detail in the supplementary materials.

We hypothesize that the lack of significant results in the separate age groups was due to interference from participants who showed a ceiling performance in the baseline phase and thus could only be shifted in one direction. We addressed this possibility in exploratory analyses discussed below.

### 3.1.1. Exploratory post-test phase analyses: excluding participants who showed a ceiling performance in the pre-test phase

One potential reason for the lack of significant results in the separate age groups, particularly the child group (for which the study was powered), is interference from ceiling performance in the pre-test phase. For instance, if a participant already produces 100% DODs in the pre-test phase, they can only shift towards higher PD (and not higher DOD) production in the post-test phase, meaning that it becomes impossible to adequately measure the effect of our main manipulation (positive or negative pre- to post-test shift in DOD production depending on bias group). While we expected no ceiling performance, some participants (21 adults and 14 children) produced exclusively PDs or exclusively DODs in the pre-test. Thus, we conducted a set of exploratory analyses including only the participants who produced both PDs and DODs in the pre-test phase, replicating the analyses discussed above. These exploratory analyses included 109 participants, 51 adults and 58 children. While this analysis is a better test of our main question, the reduction in participant size led to decreased statistical power.

### 3.1.1.1. Both age groups

The maximal frequentist model supported by the data included bias group, age group and pre-test score as fixed effects and subject and item as random intercepts with pre-test score as a random slope for item.

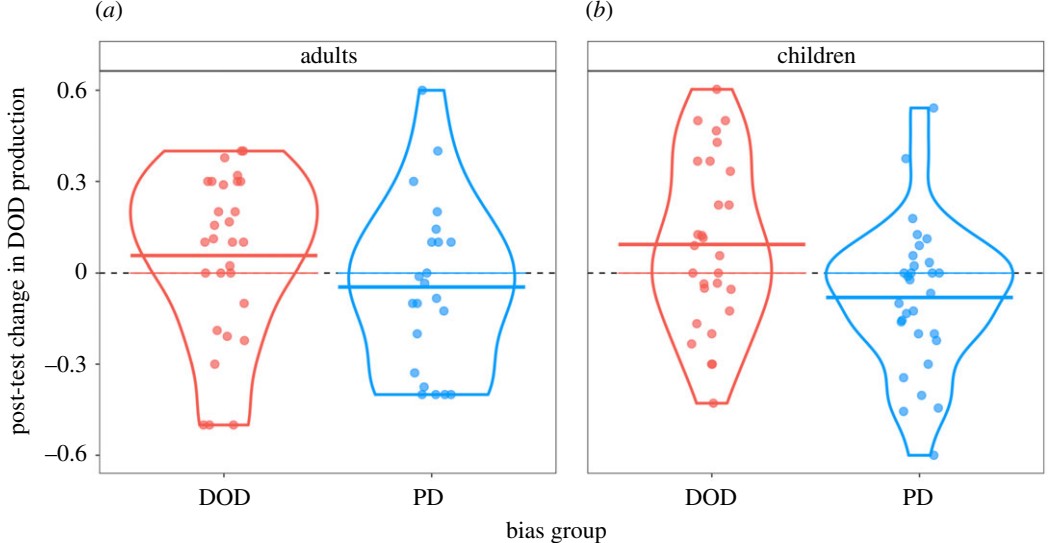

**Figure 2.** Pre- to post-test difference per age group and bias group, only including participants who did not show a ceiling performance in the pre-test. The dashed line represents no pre- to post-test change while the solid lines show the average per age and bias group shifts. For each condition, the violin lines represent the probability density of the data, and the jittered points show the pre- to post-test shift of each individual subject.

The Bayesian model included the same fixed effects with subject and item as random intercepts with bias group as a random slope for item.

We observed the expected significant bias group difference in the combined group: children showed an average 8.12%, while adults an average 3.38% pre- to post-test shift ($p = 0.018$, Bayesian posterior probability: 97.09%; see figure 2). In addition, pre-test score had a significant positive effect showing that participants with higher baseline DOD performance also produced more DODs in the post-test (all $ps < 0.001$, Bayesian posterior probabilities > 99.93%). Adults produced overall more DODs than children, but this effect did not reach significance ($p = 0.067$, Bayesian posterior probability: 96.33%). Importantly, children showed a larger pre- to post-test shift than adults, but this effect did not reach significance either ($p = 0.52$, Bayesian posterior probability: 74.08%; table 2).

### 3.1.1.2. Adult and child groups

In the adult group, the maximal frequentist model supported by the data included bias group and pre-test score as fixed effects and by-subject and by-item random intercepts with no random slopes, while the Bayesian model included the maximal effect structure. In the child group, the maximal frequentist model supported by the data included bias group, pre-test score, age in months and TROG score as fixed effects and by-subject and by-item random intercepts with by-item random slopes for pre-test score. The Bayesian analysis included the same fixed effects in addition to a by-subject random intercept with no random slopes and a by-item random intercept with random slopes for bias group. Importantly, we observed the expected pre- to post-test shift in all analyses in this dataset. Children produced 8.94% more DODs in the DOD and 7.31% more PDs in the PD bias group at post-test compared with pre-test, while adults showed a 1.96% pre- to post-test shift in the DOD and 4.81% shift in the PD bias group. The effect of bias group was significant when we analysed data from the two age groups together ($p = 0.018$, Bayesian posterior probability: 97.09%) and, this time, in the child group separately as well ($p = 0.037$, Bayesian posterior probability: 92.46%). The bias group effect did not reach significance in the adult group separately ($p = 0.36$, Bayesian posterior probability: 78.85%). In addition, pre-test score had a significant positive effect in each exploratory analysis (all $ps < 0.001$, Bayesian posterior probabilities > 99.93%).

### 3.1.2. Summary of the results in the post-test phase

In summary, the full dataset and the smaller but more representative subset dataset (that included only participants with no ceiling performance in the pre-test) both showed the expected bias group-dependent pre- to post-test shifts. While in the pre-registered analyses, the bias group difference only reached

**Table 2.** Results of the exploratory frequentist and the Bayesian analyses in the post-test phase per age group, excluding the ceiling participants. Italicized indicates significant results according to the frequentist analyses.

| comparison | frequentist | | | Bayesian | |
| --- | --- | --- | --- | --- | --- |
| | Est. [CI] | $\chi 2$ | p | mean [95% CrI] | P ($\beta > 0$) |
| **both age groups** | | | | | |
| intercept | −0.12 [−0.83, 0.58] | NA | NA | −0.13 [−0.89, 0.64] | 64.32% |
| *pre-test score* | *1.38 [0.85, 1.93]* | *28.94* | *<0.001* | *1.37 [0.86, 1.92]* | *100%* |
| *bias group* | *0.75 [0.05, 1.46]* | *5.60* | *0.018* | *0.7 [−0.02, 1.44]* | *97.09%* |
| age_group | 0.71 [−0.04, 1.52] | 3.35 | 0.067 | 0.71 [−0.06, 1.49] | 96.33% |
| bias group:age_group | −0.37 [−1.8, 0.97] | 0.41 | 0.52 | −0.46 [−1.86, 0.95] | 74.08% |
| **adult group** | | | | | |
| intercept | 0.31 [−0.38, 1.04] | NA | NA | 0.32 [−0.46, 1.09] | 80.33% |
| *pre-test score* | *1 [0.46, 1.54]* | *12.01* | *<0.001* | *1.01 [0.41, 1.65]* | *99.94%* |
| bias group | 0.4 [−0.43, 1.25] | 0.83 | 0.363 | 0.38 [−0.59, 1.32] | 78.85% |
| **child group** | | | | | |
| intercept | −0.4 [−1.44, 0.7] | NA | NA | −0.37 [−1.57, 0.8] | 26.79% |
| *pre-test score* | *2.08 [0.93, 3.2]* | *10.57* | *0.001* | *1.97 [0.83, 3.17]* | *99.98%* |
| TROG score | −0.06 [−0.71, 0.6] | 0.00 | 0.952 | −0.08 [−0.79, 0.6] | 59.17% |
| age | −0.1 [−0.79, 0.59] | 1.03 | 0.311 | −0.07 [−0.89, 0.68] | 56.69% |
| *bias group* | *1.14 [−0.15, 2.39]* | *4.35* | *0.037* | *1 [−0.36, 2.43]* | *92.46%* |
| trog score:age | −0.19 [−0.94, 0.51] | 0.04 | 0.85 | −0.22 [−0.97, 0.54] | 72.3% |
| TROG score:bias group | 0.61 [−0.73, 1.94] | 0.30 | 0.583 | 0.61 [−0.74, 2.02] | 81.15% |
| age:bias group | −1.11 [−2.57, 0.3] | 3.31 | 0.069 | −1.15 [−2.68, 0.38] | 93.3% |
| TROG score:age:bias group | −0.28 [−1.67, 1.16] | 0.16 | 0.694 | −0.21 [−1.71, 1.3] | 61.33% |

significance in the combined dataset, in the exploratory analyses both the combined and the child group separately showed a significant bias group difference. These results provide crucial initial evidence for the central claim of error-based theories that unpredictable input leads to higher rates of lasting language change than predictable input.

## 3.2. Confirming expected effects—immediate prime surprisal

**Hypothesis 1—H1a—immediate priming effects are increased if the prime structure appeared with a mismatching as opposed to a matching verb (immediate prime surprisal effect) and H1b—the immediate prime surprisal effects are larger in the child than in the adult group**

This analysis served as a manipulation check: to confirm the differences in predictability between the different bias conditions (that are designed to lead to long-term changes in the post-test phase), we assessed whether they replicated the immediate prime surprisal effects found by Peter *et al.* [27].

These analyses were carried out on the target sentences from the bias phase. The full model included as fixed effects: (i) prime structure (DOD or PD), (ii) prime-bias match (depending on whether the prime verb's bias matches or mismatches the prime structure), and (iii) age group (children or adults, in the combined model), and by-subject and by-item random intercepts and fully crossed random slopes for prime type and prime-bias match (and by-item random intercepts for age group in the combined analysis). Immediate structural priming is demonstrated if there is a greater proportion of DOD responses after DOD than PD primes, and an immediate prime surprisal effect is demonstrated if there is a significant interaction between prime-structure and prime-bias match, showing that priming effects were larger if the prime verb's bias did not match the prime structure (H1a).

In line with the prediction of error-based learning theories that error-based learning results in greater changes to children's linguistic representations than to adults', we also expected a three-way interaction between prime-structure, prime-bias match and age group, showing that the prime surprisal effect

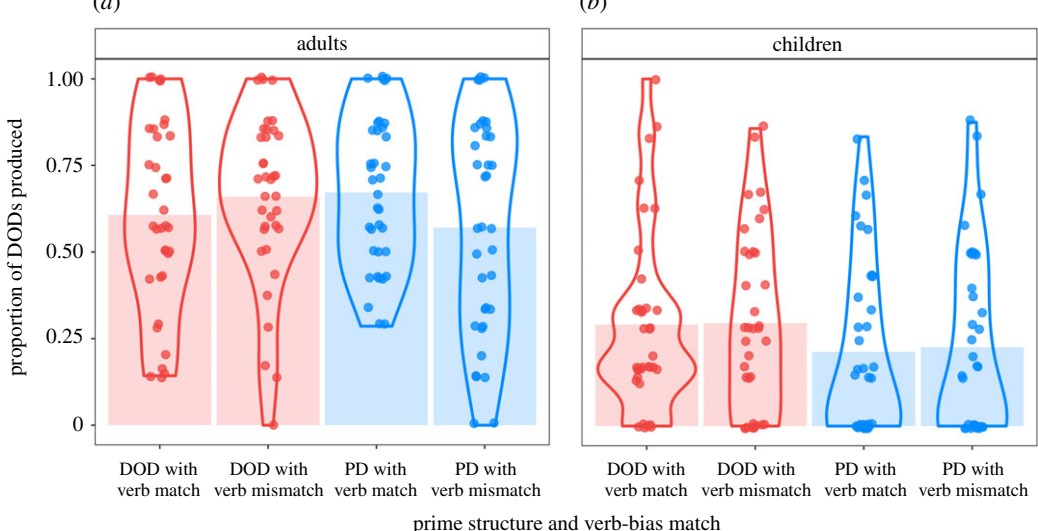

**Figure 3.** Proportion of DOD production in the bias phase by age group and condition. For each condition, the shaded bars show the mean DOD production, the violin lines represent the probability density of the data, and the jittered points show the mean DOD production levels of each individual subject averaged across all trials in the given condition.

(difference between priming after matching and mismatching verbs) is larger for the children than the adults (H1b).

### 3.2.1. Both age groups

The maximal frequentist model supported by the data included prime-structure, prime-bias match and age group as fully crossed deviation-coded fixed effects and random intercept for subject (without random slopes), plus the random intercept for item with verb match as a random slope. The Bayesian analysis featured the same fixed effects with random intercepts for subject and item and no random slopes.

The frequentist model showed a main effect of age group, indicating that, overall, adults produced more DODs than children ($p < 0.001$, Bayesian posterior probability: 100%). There was also a main effect of prime structure, suggesting that participants were more likely to produce DODs after hearing DOD than PD primes ($p = 0.015$, Bayesian posterior probability: 97.29%). Prime structure and age group produced a significant interaction ($p = 0.028$, Bayesian posterior probability: 97.29%), indicating a larger priming effect in the child than in the adult group, as predicted. In terms of prime surprisal, there was a numerically larger priming effect after mismatching (surprising) than matching (predictable) primes, but this interaction did not reach significance ($p = 0.412$, Bayesian posterior probability: 77.26%). Figure 3 suggests that a prime surprisal effect may exist in the adult, but not the child group, although the three-way interaction of prime-bias match, prime structure and age group did not reach significance ($p = 0.337$, Bayesian posterior probability: 80.84%). To explore the group-specific patterns in more detail, we carried out additional analyses on the data from the two age groups separately (figure 3; table 3).

### 3.2.2. Adult and child groups

In the adult group, the maximal frequentist model supported by the data included prime-structure and prime-bias match as fixed effects and random intercepts for subject and item with by verb-bias match random slopes for item, while the Bayesian analysis included the maximal model. In the child group, we fitted the same model used in our power calculations (§2.1), with the addition of age and syntactic knowledge as predictors. In addition to these fixed effects, the maximal frequentist model supported by the data also included a by-subject random intercept and a by-item random intercept with random slopes for prime-bias match. The Bayesian model reported here includes prime-structure and prime-bias match as fully crossed fixed effects, with TROG score and age added as covariates and with random intercepts for subject and item and random slopes for prime type and verb-bias match for both intercepts.

**Table 3.** Results of the pre-registered frequentist and the Bayesian analyses in the bias phase per age group. Italicized indicates significant results according to the frequentist analyses. NA signifies values that were not computed by the Bayesian model.

| comparison | frequentist | | | bayesian | |
|---|---|---|---|---|---|
| | Est. [CI] | $\chi^2$ | $p$ | mean [95% CrI] | $P (\beta > 0)$ |
| **both age groups** | | | | | |
| intercept | −0.4 [−0.94, 0.19] | NA | NA | −0.39 [−1.01, 0.22] | 89.75% |
| *prime type* | *0.33 [0.08, 0.57]* | *5.86* | *0.015* | *0.34 [−0.01, 0.68]* | *97.29%* |
| verb match | 0.03 [−0.26, 0.33] | 0.14 | 0.704 | 0.06 [−0.2, 0.32] | 32.7% |
| *age group* | *2.48 [1.92, 3.03]* | *69.38* | *<0.001* | *2.49 [1.94, 3.07]* | *100%* |
| prime type:verb match | −0.41 [−1.48, 0.68] | 0.67 | 0.412 | −0.41 [−0.64, 1.51] | 77.26% |
| *prime type:age group* | *−0.54 [−1.03, −0.03]* | *4.84* | *0.028* | *−0.51 [−1.03, 0.01]* | *97.29%* |
| verb match:age group | 0.22 [−0.27, 0.71] | 0.7 | 0.402 | 0.1 [−0.44, 0.62] | 63.64% |
| prime type:verb match:age group | −1.04 [−3.28, 1.15] | 0.92 | 0.337 | −0.97 [−3.2, 1.2] | 80.84% |
| **adult group** | | | | | |
| intercept | 0.84 [0.23, 1.45] | NA | NA | 0.86 [0.21, 1.51] | 99.33% |
| prime type | 0.05 [−0.27, 0.38] | 0.09 | 0.759 | 0.06 [−0.37, 0.48] | 61.40% |
| verb match | 0.16 [−0.24, 0.55] | 0.65 | 0.422 | 0.13 [−0.34, 0.6] | 71.43% |
| prime type:verb match | −0.93 [−2.3, 0.42] | 1.76 | 0.185 | −0.89 [−2.31, 0.56] | 89.03% |
| **child group** | | | | | |
| intercept | −2.09 [−2.79, −1.25] | NA | NA | −1.79 [−2.57, −1.07] | 100% |
| *prime type* | *0.97 [0.33, 1.53]* | *10.4* | *0.001* | *0.74 [0.21, 1.3]* | *99.63%* |
| verb match | 0.26 [−0.43, 0.91] | 0.19 | 0.667 | −0.04 [−0.61, 0.53] | 56% |
| age | −0.88 [−1.41, −0.25] | 1.17 | 0.279 | −0.41 [−0.86, 0.04] | 96.34% |
| *TROG* | *0.63 [0.08, 1.12]* | *5.4* | *0.02* | *0.5 [0.04, 1]* | *98.37%* |
| prime type:verb match | −1.9 [−3.86, 0.18] | 0.01 | 0.904 | −0.09 [−1.84, 1.69] | 54.08% |
| prime type:age | 0.26 [−0.39, 0.89] | 0.13 | 0.721 | NA | NA |
| verb match:age | 0.15 [−0.51, 0.79] | 0.98 | 0.322 | NA | NA |
| prime type:TROG | −0.21 [−0.79, 0.37] | 0.37 | 0.545 | NA | NA |
| verb match:TROG | −0.01 [−0.6, 0.56] | 0.28 | 0.598 | NA | NA |
| age:TROG | 0.61 [−0.03, 1.15] | 0.03 | 0.852 | NA | NA |
| prime type:verb match:age | −2.5 [−4.8, −0.08] | 0.8 | 0.372 | NA | NA |
| prime type:verb match:TROG | 0.34 [−1.77, 2.51] | 0.03 | 0.872 | NA | NA |
| Prime type:age:TROG | −0.43 [−1.02, 0.23] | 0.07 | 0.786 | NA | NA |
| Verb match:age:TROG | −0.57 [−1.17, 0.07] | 3.22 | 0.073 | NA | NA |
| *prime type:verb match:Age:TROG* | *3.57 [1.18, 5.67]* | *10.89* | *<0.001* | NA | NA |

While the frequentist model detected a significant priming effect in the child group ($p = 0.001$, Bayesian posterior probability: 99.63%), there was no evidence for priming in the adult group ($p = 0.76$, Bayesian posterior probability: 61.40%). The effect of prime surprisal (demonstrated by an interaction between prime-structure and prime-bias match) did not reach significance in either age group separately (adults: $p = 0.18$, children: $p = 0.9$). However, the pattern of responses was different in the two groups. While adults showed the expected numerically larger priming effects after surprising prime sentences, children did not show this pattern (figure 3). Furthermore, while the Bayesian posterior probability of prime surprisal was relatively high in the adult group (89.03%), it was very low in the child group (54.08%). In addition, TROG score also had a significant main effect ($p = 0.020$, Bayesian posterior probability: 98.37%) in the child group, as children with more advanced syntactic knowledge (measured by the TROG test) were more likely to produce DODs. The frequentist model also showed a significant

four-way interaction between prime-structure, prime-bias match, age and TROG score ($p < 0.001$). However, the Bayesian analysis did not include this interaction as we had to simplify the model structure due to convergence issues. This four-way interaction suggests that children who are younger and have lower TROG scores are more likely to show a sensitivity to the prime surprisal manipulation (the interaction of prime-structure and prime-bias match). We are cautious in our interpretation of this finding since there were no other significant lower level interactions in the model and we could not compute Bayesian estimates for this interaction (though it should be noted that the models were checked for overparametrization, see §3 Statistics and data analyses).

### 3.2.3. Summary of the pre-registered analyses in the bias phase

There was an immediate structural priming effect, in that participants were more likely to produce DODs after hearing DOD rather than PD primes, though this result reached significance only in the child group, not the adult group, in the age group-specific analyses. Contrary to our prediction, however, there was no immediate prime surprisal effect when both age groups were considered together, nor in the child and adult groups when considered separately (though there was a numerical prime surprisal effect in the adult group alone). Thus, neither Hypothesis 1a (replication of prime surprisal effects) nor Hypothesis 1b (larger prime surprisal effects in the child than in the adult group) were supported by the current dataset.

### 3.2.4. Exploratory bias phase analyses—including baseline DOD performance

We carried out a number of exploratory analyses to investigate these results further; all focused on determining whether our design choices could be responsible for the lack of a prime surprisal effect. Two of these analyses are reported only in the supplementary materials as they suggested that the confound proposed (bias group assignment and increasing predictability of the verb-structure pairings during the bias phase) did not affect our results. Below, we report the results of a third exploratory analysis that examined whether participants' baseline performance affected the likelihood of them showing prime surprisal effects. We repeated the analyses described above with the addition of baseline DOD performance (as measured in the pre-test phase) in the adult and the child group separately (table 4).

The maximal models included the same predictors as described in the pre-registered analyses with the addition of baseline DOD performance as a covariate. In the adult group, the maximal frequentist model supported by the data also included the random intercept for subject, and the random intercept of item with a random slope for verb-bias match. The Bayesian model contained the same fixed effects, with by-subject random intercepts with fully crossed random slopes for prime type and prime-bias match and by-item random intercepts with random intercept for verb-bias match and pre-test score. In the child group, the maximal frequentist model supported by the data included the fully crossed fixed effects of prime type, verb-bias match, age group and TROG score, with pre-test score as a covariate and by-subject and by-item random slopes and by-item random intercepts for verb-bias match. The model supported by the data in the Bayesian analysis included prime-structure and prime-bias match as fully crossed fixed effects and age, TROG score and baseline DOD performance as covariates, with by-subject and by-item random slopes, prime type and verb-bias match random slopes for both item and subject and by-item random intercepts for pre-test score.

As in the pre-registered analyses, we found a significant priming effect in the child ($p = 0.001$, Bayesian posterior probability: 99.78%), but not in the adult group ($p = 0.76$, Bayesian posterior probability: 63.43%). As expected, pre-test performance had a significant positive main effect showing that participants with high DOD production in the pre-test were also more likely to produce more DODs in the bias phase (both $ps < 0.01$, Bayesian posterior probabilities: 100%). The most important result of this analysis is that prime surprisal had a significant effect in the adult group when baseline DOD performance was included ($p = 0.049$, Bayesian posterior probability: 97.71%). While children still did not show a significant prime surprisal effect ($p = 0.36$), the posterior probability of this effect was higher in this analysis (79.42%) than in the analysis without baseline DOD performance (54.08%). As in the pre-registered child group analysis, the frequentist model also produced a significant four-way interaction between prime-structure, prime-bias match, age and TROG score ($p = 0.002$), though again this interaction was not included in the Bayesian analysis as we had to simplify the model structure due to convergence issues. As before, we are cautious in our interpretation of this finding due to the lack of corresponding lower level interactions and the absence of the Bayesian estimates.

**Table 4.** Results of the exploratory frequentist and the Bayesian analyses in the bias phase per age group. Italicized indicates significant results according to the frequentist analyses. NA signifies values that were not computed by the Bayesian model.

| comparison | frequentist | | | Bayesian | |
|---|---|---|---|---|---|
| | Est. [CI] | $\chi^2$ | p | mean [95% CrI] | $P\,(\beta > 0)$ |
| **adult group** | | | | | |
| intercept | 0.37 [−0.21, 0.97] | NA | NA | 0.39 [−0.26, 1.05] | 88.61% |
| *c_baseline* | *0.89 [0.59, 1.18]* | *29.56* | *<0.001* | *0.9 [0.55, 1.26]* | *100%* |
| prime type | 0.05 [−0.29, 0.38] | 0.1 | 0.757 | 0.06 [−0.29, 0.41] | 63.43% |
| verb match | 0.16 [−0.25, 0.54] | 0.66 | 0.416 | 0.14 [−0.33, 0.6] | 71.82% |
| *prime type:verb match* | *−1.1 [−2.18, −0.03]* | *3.89* | *0.049* | *−1.16 [−2.33, 0.02]* | *97.71%* |
| **child group** | | | | | |
| intercept | −1.43 [−2.11, −0.67] | NA | NA | −1.18 [−1.9, −0.47] | 99.94% |
| *c_baseline* | *1.08 [0.56, 1.53]* | *19.18* | *<0.001* | *1.17 [0.65, 1.72]* | *100%* |
| *prime type* | *0.97 [0.37, 1.51]* | *10.52* | *0.001* | *0.71 [0.21, 1.23]* | *99.78%* |
| verb match | 0.27 [−0.45, 0.92] | 0.21 | 0.649 | −0.05 [−0.61, 0.49] | 56.96% |
| age | −0.75 [−1.2, −0.22] | 2.77 | 0.096 | −0.42 [−0.8, −0.04] | 98.36% |
| TROG | 0.43 [−0.03, 0.84] | 3.38 | 0.066 | 0.37 [−0.02, 0.78] | 96.93% |
| prime type:verb match | −1.99 [−3.68, −0.13] | 0.84 | 0.359 | −0.61 [−2.1, 0.87] | 79.42% |
| prime type:age | 0.26 [−0.38, 0.87] | 0.11 | 0.74 | NA | NA |
| verb match:age | 0.14 [−0.51, 0.75] | 0.99 | 0.32 | NA | NA |
| prime type:TROG | −0.2 [−0.75, 0.35] | 0.37 | 0.54 | NA | NA |
| verb match:TROG | 0 [−0.52, 0.52] | 0.31 | 0.577 | NA | NA |
| age:TROG | 0.39 [−0.16, 0.85] | 0.01 | 0.929 | NA | NA |
| prime type:verb match:Age | −1.32 [−3.3, 0.72] | 0.06 | 0.805 | NA | NA |
| prime type:verb match:TROG | 0.43 [−1.31, 2.21] | 0 | 1 | NA | NA |
| Prime type:age:TROG | −0.43 [−1.02, 0.24] | 0.06 | 0.808 | NA | NA |
| verb match:age:TROG | −0.59 [−1.21, 0.1] | 3.18 | 0.074 | NA | NA |
| *prime type:verb match:age:TROG* | *2.88 [0.81, 4.66]* | *9.73* | *0.002* | *NA* | *NA* |

### 3.2.5. Summary of bias group results

In summary, we found a structural priming effect (participants were more likely to produce DODs after hearing DOD than PD primes) in both the pre-registered and in the exploratory analyses that included baseline DOD performance. However, this effect only reached significance in the child but not in the adult group. While participants overall were more likely to repeat the previously heard dative structure when it was surprising as opposed to predictable, this immediate prime surprisal effect did not reach significance in our pre-registered analyses and was only significant in the separate adult group in our exploratory analyses.

### 3.3. Additional analyses of potential interest—verb-based learning

**Hypothesis 3—H3a—verb-based long-term effects of input predictability and H3b—the shift described in H3a is stronger in the child than in the adult group**

The third set of analyses was carried out on the target sentences from the second post-test phase and the goal was to detect verb-specific long-term priming effects. Here we had to deviate from our pre-registered analyses. We mistakenly specified the inclusion of a binary verb-bias predictor (depending on whether the verb featured here as a target is overall PD- or DOD-biased) in our analyses. However, due to our between-participants design in the bias phase, verb bias and bias group are not independent predictors. Instead, to target potential verb-based learning effects, we included verb-bias

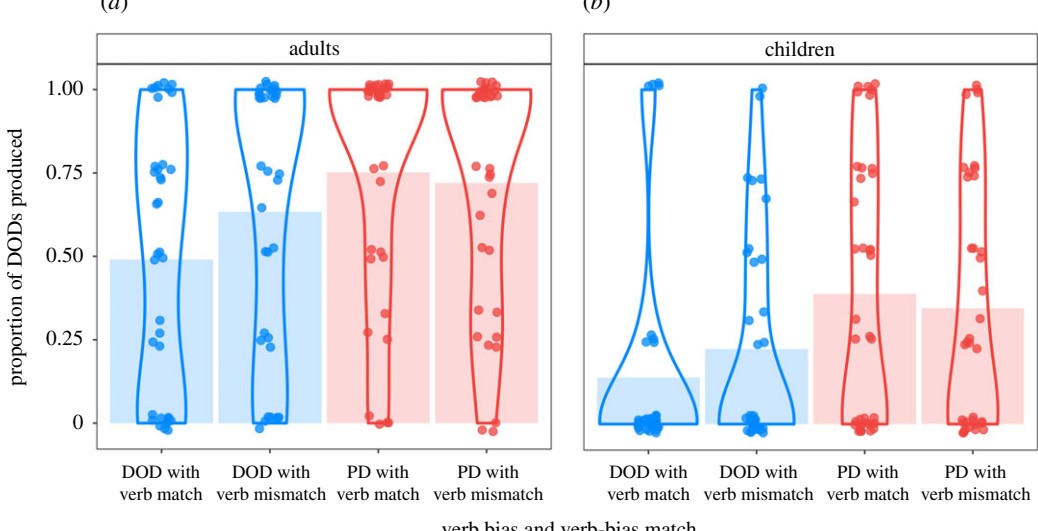

**Figure 4.** Proportion of DOD production in the second post-test phase per age group and condition. For each condition, the shaded bars show the mean DOD production, the violin lines represent the probability density of the data, and the jittered points show the mean DOD production levels of each individual subject averaged across all trials in the given condition.

match in the bias phase (match or mismatch), prime structure in the bias phase (PD or DOD) and a continuous predictor of verb bias (based on the counts from the Manchester corpus). If there is verb-based learning, we expect an interaction between verb-bias match and prime structure. The maximal models included (i) prime-structure (PD or DOD depending on which structure the verb appeared in during the bias phase), (ii) prime-bias match (depending on whether the prime verb's bias matches or mismatches the prime structure), (iii) verb-bias (based on the Manchester corpus) and (iv) age group (adults or children, in the combined analysis), and by-subject and by-item random intercepts and random slopes for prime-structure and prime-bias match in the full model.

Here, we expected to see a main effect of prime structure showing that participants are more likely to re-use the structure in which they previously heard the verb and a main effect of verb-bias, thereby replicating Peter *et al.* [27]. Crucially, a lasting verb-specific prime surprisal effect (H3a) would be demonstrated by an interaction between prime-structure and prime-bias match showing that participants are more likely to re-use the structure the target verb previously appeared in if that structure did not match the verb's bias. H3b, larger verb-based learning effects in the child then in the adult group would be demonstrated by a three-way interaction between prime-structure, prime-bias match and age group.

### 3.3.1. Both age groups together

The maximal frequentist model supported by the data included prime-structure, verb-bias match (depending on whether the verb featured here appeared in a matching or mismatching sentence), verb bias (proportion of DODs per dative occurrence in the Manchester corpus) and age group (adults or children) as fixed effects and subject as a random intercept with a random slope for verb-bias match, and a by-item random intercept with a random slope for age group. The Bayesian model included the maximal effect structure.

The frequentist model detected a significant effect of age group ($p < 0.001$, Bayesian posterior probability: 100%) and verb bias ($p < 0.003$, Bayesian posterior probability: 97.77%), suggesting that participants in the adult group were more likely to produce DODs, and that participants produced more DODs with verbs that had a higher DOD-bias in the Manchester corpus. Importantly, the main effect of prime type also reached significance ($p < 0.001$, Bayesian posterior probability: 99.80%), indicating that participants overall were more likely to produce DODs with verbs that were featured in DOD (as opposed to PD) sentences in the bias phase. However, the interaction of prime-structure and prime-bias match did not reach significance ($p = 0.98$; Bayesian posterior probability: 50.97%). Thus, we found no evidence for Hypothesis 3a—verb-based learning effects—in this analysis. Furthermore, the interaction of prime-structure, prime-bias match and age group did not reach

**Table 5.** Results of the pre-registered frequentist and the Bayesian analyses in the second post-test phase per age group. Italicized indicates significant results according to the frequentist analyses. NA signifies values that were not computed by the Bayesian model.

| comparison | frequentist | | | Bayesian | |
|---|---|---|---|---|---|
| | Est. [CI] | $\chi^2$ | $p$ | mean [95% CrI] | $P(\beta > 0)$ |
| **both age groups** | | | | | |
| intercept | −0.23 [−0.85, 0.4] | NA | NA | −0.24 [−0.92, 0.44] | 76.56% |
| *verb bias* | *0.96 [0, 1.86]* | *8.81* | *0.003* | *0.99 [0.03, 1.97]* | *97.77%* |
| *age group* | *3.75 [2.13, 5.07]* | *34.34* | *<0.001* | *3.77 [2.55, 5.14]* | *100%* |
| verb match | −0.41 [−0.84, 0.01] | 1.94 | 0.164 | −0.45 [−1.08, 0.18] | 92.64% |
| *prime type* | *0.81 [0.36, 1.27]* | *13.94* | *<0.001* | *0.91 [0.3, 1.58]* | *99.8%* |
| age group:verb match | 0.17 [−0.68, 1.03] | 0.15 | 0.7 | 0.35 [−0.8, 1.55] | 71.66% |
| age group:prime type | −0.04 [−0.88, 0.77] | 0.01 | 0.932 | 0.03 [−1.14, 1.22] | 52.22% |
| verb match:prime type | −0.07 [−4.19, 3.81] | 0 | 0.984 | −0.06 [−4.09, 3.97] | 50.97% |
| age group:verb match:prime type | −0.15 [−4.97, 4.67] | 0 | 0.952 | −0.47 [−5.13, 4.24] | 58.18% |
| **adult group** | | | | | |
| intercept | 1.87 [−0.07, 3.26] | NA | NA | 0.87 [0.52, 1.28] | 100% |
| *verb bias* | *1.29 [−0.21, 2.58]* | *7.07* | *0.008* | *0.49 [−0.34, 1.3]* | *89.19%* |
| *prime type* | *0.83 [0.19, 1.45]* | *7.7* | *0.006* | *0.45 [−0.02, 0.93]* | *96.84%* |
| verb match | −0.33 [−0.95, 0.3] | 1.18 | 0.278 | −0.11 [−0.59, 0.36] | 68.71% |
| prime type:verb match | −0.56 [−7.4, 5.13] | 0.03 | 0.852 | −0.72 [−4.04, 2.45] | 68.29% |
| **child group** | | | | | |
| intercept | −2.39 [−3.42, −0.84] | NA | NA | −2.12 [−3.13, −1.26] | 100% |
| *verb bias* | *0.84 [−0.24, 1.8]* | *7.32* | *0.007* | *0.86 [−0.17, 1.87]* | *95.26%* |
| *prime type* | *0.68 [−0.51, 1.78]* | *4.97* | *0.026* | *0.89 [0.1, 1.72]* | *98.68%* |
| verb match | −0.58 [−1.72, 0.69] | 2.23 | 0.136 | −0.66 [−1.54, 0.2] | 93.17% |
| ageTR | −0.56 [−1.49, 0.59] | 0.03 | 0.866 | NA | NA |
| TROG | 0.57 [−0.5, 1.41] | 2.61 | 0.106 | NA | NA |
| prime type:verb match | −1.59 [−7.33, 4.18] | 0.02 | 0.901 | 0.56 [−3.93, 5.12] | 59.96% |
| prime type:age | −0.23 [−1.31, 0.84] | 0.02 | 0.875 | NA | NA |
| verb match:age | −0.35 [−1.42, 0.78] | 0.49 | 0.483 | NA | NA |
| *prime type:TROG* | *0.81 [−0.33, 1.8]* | *4.12* | *0.042* | *NA* | *NA* |
| verb match:TROG | −0.35 [−1.38, 0.74] | 1.16 | 0.281 | NA | NA |
| age:TROG | 0.88 [−0.35, 1.85] | 0.99 | 0.321 | NA | NA |
| prime type:verb match:age | −2.34 [−6.33, 2.01] | 0.58 | 0.445 | NA | NA |
| prime type:verb match:TROG | −1.17 [−4.88, 3.01] | 0.8 | 0.372 | NA | NA |
| prime type:age:TROG | −0.01 [−0.93, 0.92] | 0.01 | 0.92 | NA | NA |
| verb match:age:TROG | 0.28 [−0.78, 1.31] | 0.47 | 0.49 | NA | NA |
| *prime type:verb match:age:TROG* | *3.96 [−0.58, 7.7]* | *5.05* | *0.025* | *NA* | *NA* |

significance either ($p = 0.95$, Bayesian posterior probability: 58.18%), providing no evidence for Hypothesis 3b, larger verb-based learning effects in the child than in the adult group (figure 4; table 5).

### 3.3.2. Adult and child group

In the adult group, the maximal frequentist model supported by the data included prime-structure, verb-bias match and verb bias as fixed effects and by-subject and by-item random intercepts with verb-bias

random slopes for subject. The Bayesian model included the same fixed effects and by-subject and by-item random intercepts and by-subject random slopes for prime type and by-item random slopes for verb-bias match. The maximal frequentist model supported by the data in the child group included prime-structure, verb-bias match, verb bias, age and TROG score as fixed effects and by-subject and by-item random intercepts without random slopes. The Bayesian model supported by the data included prime-structure, verb-bias match and verb bias as fixed effects and by-subject and by-item random intercepts and by-subject random intercepts for verb-bias match and by-item random slopes for prime type.

The pattern of results was similar in the two age groups. The main effects of prime structure (adults: $p = 0.006$, Bayesian posterior probability: 96.84%; children: $p = 0.026$, Bayesian posterior probability: 98.68%) and verb bias (adults: $p = 0.008$, Bayesian posterior probability: 89.19%; children: $p = 0.007$, Bayesian posterior probability: 95.26%) were significant, showing that participants were more likely to produce a DOD structure if they heard the target verb in a DOD structure in the bias phase and if the verb was more DOD-biased. In the child group, this structure repetition was significantly stronger in children with higher TROG scores ($p = 0.042$, this interaction was not included in the Bayesian analyses). Importantly, the interaction of prime type and prime-bias match did not reach significance in the separate age groups either (both $ps > 0.85$). The Bayesian analysis suggested that the posterior probability of a verb-based learning effect is 68.29% in the adult and 59.96% in the child group. Despite the lack of evidence for verb-based learning effects, the frequentist model in the child group showed a significant four-way interaction between prime-structure, prime-bias match, age and TROG score ($p = 0.025$; this interaction was not included in the Bayesian models as we had to simplify the model structure due to convergence issues). This four-way interaction suggests that younger children who have a lower TROG score are more likely to show sensitivity to our verb-dependent error-based learning measure (the interaction of prime-structure and verb-bias match in the bias phase). As with the four-way interactions discussed in the bias phase, we are cautious in our interpretation of this finding due to the lack of most corresponding lower level interactions and the absence of the Bayesian estimates.

### 3.3.3. Summary of the results of the second post-test phase

While both age groups were more likely to re-use the dative structure they heard the verbs with in the bias phase, this effect was not modulated by whether the structure was surprising or predictable. This study therefore provided no evidence for Hypothesis 3a, verb-specific error-based learning effects. As we found no significant interaction between prime type, verb-bias match and age group, this analysis did not support Hypothesis 3b (larger verb-based learning effects in the child than in the adult group) either (table 6).

## 3.4. Discussion

The goal of the current study was to evaluate the central prediction of error-based theories of language acquisition: that surprising linguistic input leads to higher rates of learning than predictable input. To achieve this, we embedded a prime surprisal study in a four-stage intervention study to assess both the short- and long-term effects of predictability.

The most important result of the study was that we found support for the above claim. In the first (pre-test) phase of the study, we assessed participants' baseline rates of dative production. In the second (bias) phase, we presented participants with surprising and unsurprising sentences, designed to bias them towards one of the dative structures. In the third (post-test) phase, we reassessed participants' spontaneous rates of dative production. As expected, we found that both adults and children showed an accelerated learning rate for the same structure if it was previously presented in a surprising as opposed to predictable context. Both age groups were more likely to produce DODs in the DOD- as opposed to the PD-bias group in the post-test phase. Furthermore, both adults' (average 4.25% shift) and children's (average 6.12% shift) pre- to post-test production shifted towards the dative structure they were exposed to in surprising sentences in the previous phase. This effect (difference between DOD- and PD-bias group, with baseline DOD production taken into account) was significant in the pre-registered analyses that included both the adult and the child group. In addition, although the frequentist analysis did not reach significance in either the adult or child groups separately, the Bayesian analysis suggested that the posterior probability of these effects was high, especially in the child group (adult group: 82.77%, child group: 91.22%). In sum, even though

**Table 6.** Appearance of expected response patterns per study phase and age group. In the bias phase the table shows whether participants demonstrated immediate prime surprisal and structural priming effects, the latter in brackets. In the post-test phase, the table shows whether participants showed more learning for abstract structures after surprising as opposed to predictable sentences. In the second post-test phase, the table shows whether verb-based learning rates were higher in surprising sentences. In brackets we can see whether participants were likely to use the dative structure in phase 4 that specific verbs appeared with in the bias phase.

| | age group | | |
| --- | --- | --- | --- |
| | adults | children | both groups together |
| bias phase—prime surprisal (priming) | ✓ (✗) baseline DOD production included | ✗ (✓) | ✓ (✓) Prime surprisal n.s. |
| post-test phase—abstract learning | ✓ n.s. | ✓ no ceiling in pre-test | ✓ |
| second post-test phase—verb-based learning (structure repetition) | ✗ (✓) | ✗ (✓) | ✗ (✓) |

participants in both groups heard the same number of DOD and PD structures, their production changed based on which structure was predictable and which one was surprising in the previous phase. This is crucial evidence for a central prediction of the Dual-path model [11]: an increased learning rate for the same structure when it appeared earlier in a surprising as opposed to a predictable sentence.

As the magnitude of the pre- to post-test shift in the child group was similar to what we estimated in our power calculations (average 5% shift in the power calculation and an average 6.12% shift in the child dataset), the lack of significant effects when we analysed the child group data alone was surprising. We surmised that this might have been due to ceiling performance: 14 children and 21 adults produced only DODs or only PDs in the pre-test phase. Thus, we carried out a set of exploratory analyses excluding participants with ceiling pre-test performance, which demonstrated significant learning effects when we analysed data from the two age groups together and, this time, in the child group separately as well.

To our knowledge, this is the first study that has found such learning effects, providing initial experimental evidence for a central claim of error-based learning theories: that surprising input leads to more learning compared with predictable input. These results are also in line with previous studies demonstrating that children's production frequencies can be shifted towards a less frequent structure by exposure in the bias phase [30,34]. Our study contributes to this literature by specifically showing that these differences can be traced back to input predictability.

A secondary goal of the study was to determine if we could replicate the immediate prime surprisal effects found in previous studies [26,27]. We found larger priming effects after surprising as opposed to predictable primes in the adult group, but neither the priming nor the prime surprisal effect reached significance in the pre-registered analyses. The Bayesian analysis showed that the posterior probability of priming was 61.40% while the posterior probability of prime surprisal was 89.03%. Children showed a significant priming effect (Bayesian posterior probability: 99.63%), but there was no sign of prime surprisal in this group (Bayesian posterior probability: 54.08%).

This failure to replicate immediate structural priming and prime surprisal effects was unexpected, so we explored potential explanations in exploratory analyses. Our study had a between-subjects design in the bias phase, where participants either heard only DOD-biased verbs (paired with either DOD (predictable) or PD (surprising) structures), or only PD-biased verbs (paired with either PD (predictable) or DOD (surprising) structures), table 1. This is unlike previous prime surprisal studies in which all participants heard all four types of sentences. While this design was necessary to contrast learning rates for predictable versus surprising sentences and thereby test our primary hypothesis, it may have interfered with any immediate prime surprisal effects. Two of these analyses (reported in the electronic supplementary material) did not change the pattern of results. The third analysis in which we included baseline DOD performance in the models, to control for the effect of participants'

individual differences in baseline performance, was reported above. Here, adults showed significant prime surprisal effects when their baseline DOD production rate was taken into account (Bayesian posterior probability: 97.71%), although there was no significant prime surprisal effect in the child group (Bayesian posterior probability: 79.42%). The Bayesian posterior probability of a prime surprisal effect was higher in the child group when baseline dative production was taken into account (79.42% as opposed to 54.08%). It is possible that, due to the large variability in children's DOD production, prime surprisal in childhood is particularly sensitive to the difference between within and between-participant designs. (Note though that these exploratory analyses may have been underpowered as our power calculation did not include the additional baseline predictor.)

While we cannot be certain of the source of the discrepancy between long-term learning effects and immediate prime surprisal in the child group, it is worth noting that the pattern we observed is not compatible with the Dual-path model. This model suggests that immediate priming effects are the product of the same learning mechanism that leads to long-lasting changes in syntactic knowledge. It would therefore predict similar effects with respect to immediate prime surprisal and learning. The disconnect between these effects raises questions about whether learning and priming are always induced by the same mechanism. However, as the main goal of our study was not to assess the relationship between immediate prime surprisal and long-term learning, the results of this comparison must be interpreted with caution and followed up in further studies.

The last phase of the study, the second post-test, targeted verb-dependent error-based learning effects. In this phase, we expected participants to be more likely to use the same dative structure that specific verbs appeared with in the bias phase. We also expected that the likelihood of structure repetition would be higher if the structure was unexpected in the bias phase. While we found that participants in both age groups were significantly more likely to repeat the structures the verb appeared with previously, this effect was not modulated by how surprising the structure was (it did not reach significance in the frequentist analyses and the Bayesian analyses suggested that the posterior probability of this effect is between is 50.97% and 68.29% depending on age group). This study therefore does not provide evidence for verb-dependent error-based learning effects. At first glance, these results seem to be in conflict with unpublished results from Fisher & Lin [37] who detected stronger verb-based learning effect after unpredictable verb-structure pairings. However, it is not possible to draw strong conclusions based on the absence of these effects in the current study, since the final phase of the study was exploratory and provided a less sensitive test of learning than the main test of abstract learning. The partially between-participant design led to both uneven target verb-bias rates and uneven baseline DOD rates in the different conditions that may have masked any learning effects. Furthermore, both participants' abstract learning effects and their previous dative production with the same verbs may have interfered with the results in this phase.

Finally, we assessed whether the priming and learning effects we found were sensitive to age and syntactic knowledge. The Dual-path model predicts that learning effects should decrease as the learner accumulates more linguistic knowledge and develops stronger linguistic representations. In our study, this prediction would be supported if children consistently showed both larger learning and prime surprisal effects than adults (H1b, H2b and H3b) and if, within the child group, these effects were larger in children who were younger or had less advanced syntactic knowledge. The current study did not find any conclusive evidence for any such effects. While both the abstract (post-test phase) and verb-based learning effects (second post-test phase) were numerically larger in the child than in the adult group, the interaction of learning effect and age group did not reach significance in any of our analyses. In addition, there was no significant effect of immediate prime surprisal in the child group during the bias phase.

The contribution of age and syntactic knowledge (measured by the TROG test) also did not lead to a clear conclusion. As suggested by the Dual-path model, younger children and those with lower TROG scores showed larger learning and prime surprisal effects in most analyses, except in the post-test phase, where TROG score had a positive effect. However, none of these effects reached significance. Despite the lack of lower level interactions, the frequentist models detected a significant interaction of immediate prime surprisal age and TROG score and verb-based learning, age and TROG score, indicating that younger children with lower TROG scores demonstrate larger prime surprisal and verb-based learning effects. While these results are in line with the predictions of the Dual-path model, in the absence of lower level interactions and Bayesian estimates, we cannot be certain that these effects are reliable. Our study therefore does not provide conclusive evidence regarding the contribution of age and syntactic knowledge to the learning mechanism in question, and this question needs to be addressed in future studies which are designed (and adequately powered) for exploring these comparisons.

The current study had three main limitations. First, as our main interest was surprisal-dependent abstract learning, we had to induce different levels of predictability for the different dative structures, which led to compromises when designing phases targeting immediate prime surprisal and verb-based learning. We have discussed these modifications and their potential consequences in the previous sections. Second, as we targeted a previously untested question, we had to base our power calculations on effects corresponding to similar, but not identical research questions. As a result, we were unable to account for all the factors that emerged. Thus, it is crucial for future studies to replicate our results using power calculations that are updated based on the current data. The final limitation of our study lies in the nature of our method, the prime surprisal paradigm. While it can directly address potential changes in language production depending on the predictability of the input, it does not give us any information about online processing differences between predictable and surprising sentences. Future work should therefore combine this method with online measures such as EEG or eye-tracking in order to explore how these learning effects unfold over time.

## 4. Conclusion

Our study embedded the prime surprisal paradigm in a four-stage intervention study to address a strong, but as yet not directly tested, claim of error-based learning theories that surprising input leads to more learning than predictable input. Although we did not replicate all the results from the previous literature (in particular, the lack of immediate prime surprisal in our child group was unexpected), we confirmed our primary hypothesis: that less predictable (more surprising) input leads to higher rates of lasting syntactic representational change compared with predictable input. Both adults' and children's dative production shifted towards the (surprising) structure they were biased towards in the previous phase. To our knowledge, this is the first demonstration that exposure to the same syntactic structure leads to an increased learning rate if this structure was presented in a context that made it surprising rather than predictable. The present work also contributes by establishing an experimental paradigm that can be used to target further aspects of error-based learning theories of language acquisition in the future.

Ethics. The study has received ethical approval from the IPHS Research Ethics Committee (ethics approval reference: IPHS-1617-SMC-273-Generic- PSYC 1011-111). All participants or their caregivers gave informed consent before taking part in the study.

Data accessibility. All datasets, codes, testing logs and other digital research materials are deposited at the Dryad Digital Repository: https://dx.doi.org/10.5061/dryad.3n5tb2rdq [58].

Authors' contributions. J.F., J.P. and C.R. contributed to the study conception and design, and A.J. was responsible for carrying out the power calculations and statistical analyses. All four authors contributed to the critical revisions, and J.F. drafted the manuscript. All authors gave final approval for publication.

Competing interests. The authors have no competing interests.

Funding. J.F.'s work was enabled by an ESRC PhD Studentship. J.P., C.R. and A.J. received no grants or funding in support of this project.

Acknowledgements. We would like to thank Holly Branigan and Michelle Peter for inspiring and aiding this project from the start and Ben Ambridge, Kate Messenger and the LaDD group for providing feedback. We would also like to thank Beth Gerrard for her invaluable help in recruiting participants and collecting the data. Last but not least we would like to thank all participating children and adults who played Bingo and their parents and schools for facilitating it.

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
