## [Reviewer comments · Royal Society Open Science]

Review History

Decision letter (RSOS-180455.R0)

23-Mar-2018

Dear Ms Fazekas,

I write you in regards to manuscript RSOS-180455 entitled "Do children learn from their mistakes? Evaluating error-based theories of language acquisition" which you submitted to Royal Society Open Science.

We routinely triage submissions for scientific soundness, clarity and general adherence to the Registered Reports guidelines. For submissions that have promise but are not yet suitable for in-depth Stage 1 review, we offer feedback to help authors maximise the chances that reviewers will respond positively to a resubmission.

We have concluded that your submission is not yet suitable for in-depth review and has therefore been rejected at this time, but we believe it will be suitable once several issues are addressed. We therefore invite a resubmission. Further comments from the Associate Editor may be found at the end of this letter.

If you wish to revise your manuscript in light of the below comments please submit your manuscript as a new submission and mention this previous manuscript ID in your covering letter. You should also provide a detailed response to the below comments in the cover letter.

Please note that Royal Society Open Science will introduce article processing charges for all new submissions received from 1 January 2018. Registered Reports submitted and accepted after this date will ONLY be subject to a charge if they subsequently progress to and are accepted as Stage 2 Registered Reports. If your manuscript is submitted and accepted for publication after 1 January 2018 (i.e. as a full Stage 2 Registered Report), you will be asked to pay the article processing charge, unless you request a waiver and this is approved by Royal Society Publishing. You can find out more about the charges at <http://rsos.royalsocietypublishing.org/page/charges>. Should you have any queries, please contact openscience@royalsociety.org.

Thank you for considering Royal Society Open Science for the publication of your registered report.

Kind regards,
Andrew Dunn
Royal Society Open Science
openscience@royalsociety.org

on behalf Chris Chambers (Registered Reports Editor, Royal Society Open Science)
openscience@royalsociety.org

Associate Editor Comments to Author:

Associate Editor

Comments to the Author:

1. Please ensure that there is a direct correspondence between the proposed hypotheses, the power analysis/analyses, and the proposed statistical tests in every case. At present these links not sufficiently clear. The main power analysis reported on p14 appears to target one interaction in the mixed model but the hypotheses reported later involve a range of effects and interactions. Please ensure that the power analysis maps directly and precisely on to each of the proposed hypothesis tests (including the manipulation checks), each of which – in turn – needs to correspond to a specified hypothesis. It would be helpful also for reviewers if you could supply the code used to perform the simulations underpinning your power analysis (e.g. depositing this on the OSF and linking to this URL in the Stage 1 manuscript).

2. On page 13 you note that "We made conservative estimates of these shifts based on effects in previous studies showing immediate prime surprisal effects and studies showing a long-term shift in children's production frequencies of abstract syntactic structures in similar age groups." Please provide full details of how the effect size was determined, justifying why it is the smallest effect size of interest.

3. In section 2.7, please clarify whether coders will be blinded to the experimental conditions.

Author's Response to Decision Letter for (RSOS-180455.R0)

Dear Dr. Dunn,

Please find an updated version of the previous manuscript in which we have addressed the editorial comments the following ways.

1. We have provided more information on our power analyses (clarifying the correspondence between proposed hypotheses and power analyses, and giving OSF links to the power analyses codes in the manuscript).
2. We have provided full details on how the smallest effect sizes of interest for the power analyses were calculated
3. We have clarified whether coders will be blinded to the experimental conditions.
4. We have also revised our analysis strategy for our key comparison in order to achieve a more powerful statistical approach that maps more directly to our key comparison of interest.

Yours sincerely,
Judit Fazekas
University of Liverpool
(for all co-authors)

RSOS-180877.R0

Review form: Reviewer 1

Do you have any ethical concerns with this paper?

No

Have you any concerns about statistical analyses in this paper?

No

Recommendation?

Accept with minor revision

Comments to the Author(s)

To the authors

The authors examine whether surprising (less predictable) input leads to more lasting changes in language than predictable input using a novel approach called- prime surprisal. Despite my trivial concerns, I feel the review is adequate and motivates the research questions. The method is meticulously planned to address the research questions and the study warrants registered report route of investigation. While, I feel the design is already sufficient, I raise some points which the authors may want to clarify before the protocol is accepted in principle.

Introduction

Specific (must address)

1. Page 2-3: It is not clear to me what authors mean by 'Second, rather than simply seeking to define children's state of knowledge at different developmental stages, these models explain how children move from one knowledge state to another and hence provide a specific learning mechanism that can be tested experimentally'. An example (and a reference, if suitable) would make the reading easier.

2. Page 4: Lines 38 onwards: Authors argue that previous findings may have been confounded with the process of integration. I am not sure, how their present design overcomes this flaw.

3. Page 7: Lines 33:35. Theories (and authors) argue that the error signal leads to learning. For instance, when the child hears a violating (DOD with 'bring') phrase, the child generates an error signal. This assumes that the child is aware of what the usual occurrence pattern is, that is, DOD with 'give'. So, for this design to work, child must have already acquired/mastered this knowledge (the end product of learning?)

If authors indeed agree with my above argument, I would like to question the error based theories' significance in language acquisition research. That is, for it to work you need (a well developed ?) language in place. Authors are requested to clarify this (in paper if it is sensible or to me if my argument is baseless). For me this is a serious question, because, if authors agree with me, the present study could only answer 'whether error based priming surprisal effect exist / demonstrable' but NOT to the bigger question of whether error based learning is crucial for (language) learning.

Or

This is where the comparison with adults (fully developed system) and children (malleable system) is justified? [see my point 3 & 4 in Method]

General

1. Is this 'error based learning' mechanism unique language learning mechanism? Or is it proposed as a general learning mechanism that is present across animal species?

Method

Specific

1. Is there any reason why authors want the experimenter to describe (the primes) and not use a pre-recorded description? A recorded material brings uniformity between trials and participants (The voice of the experimenter is prone to mood/emotional changes). However, I do believe that authors have a reason to opt for this approach. It may be worth explaining in the text.

2. Page 10: Line 52, Should this '(i.e. DODs for participants in group A)' read as '(i.e., PD's for participants in group A)'?? Or did I get it wrong. Please clarify!

3. Must justify more strongly the inclusion of adults in the design? Is it a reference sample to validate the design?.Page 13: Line 26: 'These age groups have shown sensitivity to verb-bias manipulations both in the target verb and in the prime verb (prime surprisal) conditions in a priming study involving dative structures'. Then why have adults? The results may be easier to interpret if we EXCLUDE adults from the analysis.

4. Page 20: Lines 37-40 'but, in order to explore the group-specific patterns in more detail, we will also carry out analyses on the data from the two age groups separately.' Why is this necessary? Do the authors have differential predictions for these groups that are theoretically motivated? Without that stated, I assume that the authors have little interest in adult's group. In which case, I suggest the authors to drop the adults from the plan and focus on children. Or if they do feel adults are needed they must justify the inclusion in the manuscript. If their justification is weak, I would like to only have a small adult sample (preliminary, exploratory) and see if they see the effects and then take it to children.

Note: I am not insisting on the changes in 3 & 4. I feel the authors have some assumptions that need explicitly stating in the manuscript. I leave it to authors convenience to work it out or stay with the same design if they could justify having adults.

5. Page 15, Design section. I would like to see the predictions stated right in this section for ease of following.

6. Given the argument that error based learning might underlie language acquisition, would it be nice to have a language measure and see if participant's ability to generate error signal/ or learn from the surprisal effect predict their language proficiency?

7. Page 18: Line 26: Check the bracket usage

8. Since the aim is to examine whether the learning that results is 'lasting', shouldn't there be another post-test phase after a delay? Could the present post-test be broken in to two? (immediate and delayed).

Decision letter (RSOS-180877.R0)

30-Aug-2018

Dear Ms Fazekas

On behalf of the Editors, I am pleased to inform you that your Stage 1 Manuscript RSOS-180877 entitled "Do children learn from their mistakes? Evaluating error-based theories of language acquisition" has been accepted in principle for publication in Royal Society Open Science subject to minor revision in accordance with the referee and editor suggestions. Please find their comments at the end of this email.

The reviewers and handling editors have recommended publication, but also suggest some minor revisions to your manuscript. Therefore, I invite you to respond to the comments and revise your manuscript.

Full author guidelines can be found here
<http://rsos.royalsocietypublishing.org/content/registered-reports>.

Please note that Royal Society Open Science charge article processing charges for all new submissions that are accepted for publication. Charges will also apply to papers transferred to Royal Society Open Science from other Royal Society Publishing journals, as well as papers submitted as part of our collaboration with the Royal Society of Chemistry (<http://rsos.royalsocietypublishing.org/chemistry>). If your manuscript is newly submitted and subsequently accepted for publication, you will be asked to pay the article processing charge, unless you request a waiver and this is approved by Royal Society Publishing. You can find out more about the charges at <http://rsos.royalsocietypublishing.org/page/charges>. Should you have any queries, please contact openscience@royalsociety.org.

on behalf of Chris Chambers (Subject Editor, Royal Society Open Science)
 openscience@royalsociety.org

Associate Editor Comments to Author (Chris Chambers):

At the outset, please accept my apologies for the slow handling of your submission, which does not meet the standards of promptness that we seek maintain at RSOS. The reason for the delay was that we experienced significant problems finding reviewers over the summer break, compounded by the fact that after receiving one high-quality review very early (Reviewer 1 below), a second reviewer became non-responsive. As you know, I had forwarded you a decision based on one review via email; however today we received the belated second review; therefore this decision letter is now based on the two reviews.

The good news is that your manuscript was positively appraised by the reviewers. Please note carefully their recommendations concerning clarity of the Introduction, rationale for the hypotheses and current design, provision of additional methodological detail, and concerns regarding the statistical analysis. Provided you are able to respond thoroughly to these comments, I would anticipate IPA to be forthcoming without requiring further in-depth review.

Reviewer comments to Author:

Reviewer: 1

Comments to the Author(s)

To the authors

The authors examine whether surprising (less predictable) input leads to more lasting changes in language than predictable input using a novel approach called- prime surprisal. Despite my trivial concerns, I feel the review is adequate and motivates the research questions. The method is meticulously planned to address the research questions and the study warrants registered report route of investigation. While, I feel the design is already sufficient, I raise some points which the authors may want to clarify before the protocol is accepted in principle.

Introduction

Specific (must address)

1. Page 2-3: It is not clear to me what authors mean by 'Second, rather than simply seeking to define children's state of knowledge at different developmental stages, these models explain how children move from one knowledge state to another and hence provide a specific learning mechanism that can be tested experimentally'. An example (and a reference, if suitable) would make the reading easier.
2. Page 4: Lines 38 onwards: Authors argue that previous findings may have been confounded with the process of integration. I am not sure, how their present design overcomes this flaw.
3. Page 7: Lines 33:35. Theories (and authors) argue that the error signal leads to learning. For instance, when the child hears a violating (DOD with 'bring') phrase, the child generates an error signal. This assumes that the child is aware of what the usual occurrence pattern is, that is, DOD with 'give'. So, for this design to work, child must have already acquired/mastered this knowledge (the end product of learning?)

If authors indeed agree with my above argument, I would like to question the error based theories' significance in language acquisition research. That is, for it to work you need (a well developed ?) language in place. Authors are requested to clarify this (in paper if it is sensible or to me if my argument is baseless). For me this is a serious question, because, if authors agree with me, the present study could only answer 'whether error based priming surprisal effect exist /

demonstrable' but NOT to the bigger question of whether error based learning is crucial for (language) learning.

Or

This is where the comparison with adults (fully developed system) and children (malleable system) is justified? [see my point 3 & 4 in Method]

General

1. Is this 'error based learning' mechanism unique language learning mechanism? Or is it proposed as a general learning mechanism that is present across animal species?

Method

Specific

1. Is there any reason why authors want the experimenter to describe (the primes) and not use a pre-recorded description? A recorded material brings uniformity between trials and participants (The voice of the experimenter is prone to mood/emotional changes). However, I do believe that authors have a reason to opt for this approach. It may be worth explaining in the text.

2. Page 10: Line 52, Should this '(i.e. DODs for participants in group A)' read as '(i.e., PD's for participants in group A)'? Or did I get it wrong. Please clarify!

3. Must justify more strongly the inclusion of adults in the design? Is it a reference sample to validate the design?.Page 13: Line 26: 'These age groups have shown sensitivity to verb-bias manipulations both in the target verb and in the prime verb (prime surprisal) conditions in a priming study involving dative structures'. Then why have adults? The results may be easier to interpret if we EXCLUDE adults from the analysis.

4. Page 20: Lines 37-40 'but, in order to explore the group-specific patterns in more detail, we will also carry out analyses on the data from the two age groups separately.' Why is this necessary? Do the authors have differential predictions for these groups that are theoretically motivated? Without that stated, I assume that the authors have little interest in adult's group. In which case, I suggest the authors to drop the adults from the plan and focus on children. Or if they do feel adults are needed they must justify the inclusion in the manuscript. If their justification is weak, I would like to only have a small adult sample (preliminary, exploratory) and see if they see the effects and then take it to children.

Note: I am not insisting on the changes in 3 & 4. I feel the authors have some assumptions that need explicitly stating in the manuscript. I leave it to authors convenience to work it out or stay with the same design if they could justify having adults.

5. Page 15, Design section. I would like to see the predictions stated right in this section for ease of following.

6. Given the argument that error based learning might underlie language acquisition, would it be nice to have a language measure and see if participant's ability to generate error signal/ or learn from the surprisal effect predict their language proficiency?

7. Page 18: Line 26: Check the bracket usage

8. Since the aim is to examine whether the learning that results is 'lasting', shouldn't there be another post-test phase after a delay? Could the present post-test be broken in to two? (immediate and delayed).

Reviewer comments to Author:

Reviewer: 2

Comments to the Author(s)

To the authors

The registered report presents an interesting idea for a study that is very likely to advance our understanding of the mechanisms underlying language acquisition. The study design is clear; the method seems appropriate and is well motivated by previous research. The number of

participants is sufficient and supported by power calculations. The hypotheses are clearly formulated and plausible.

My suggestions for the authors concern two issues: the statistical analysis and the framing. In the proposal, the authors write that they plan to conduct mixed effects analyses. In fact, they intend to “fit maximal models and simplify the random effects structure until the model converges” (p. 20). This does not seem to be the right practice. Note that model non-convergence is not the same as a given predictor not having an effect. The paper by Barr et al. (2013, JML) suggests to simplify the model structure when predictors do not show an effect and subsequently compare the slimmer model with the previous one to check whether the adjusted model does a better job at explaining the data. I gather that the authors are aware that using mixed-effect models on categorical data often leads to model non-convergence, an observation also reported by Barr and colleagues: “[...] practical experience with fitting LMEMs to datasets with categorical response variables, in particular with mixed logit models, suggests more difficulty in getting maximal models to converge” leading them to conclude that “although our arguments and recommendations [...] still apply in principle, they might need to be modified for non-continuous cases” (see Baayen et al., 2017, for further discussion). In light of the lack of a clear standard for mixed-effects analyses on categorical data I urge the authors to complement their analyses using other techniques, i.e. magnitude estimation (Cumming, 2014) or Bayesian approaches (Bolker et al., 2009, TiEE, for further discussion).

Moreover, to my taste, the description of the role/importance of prediction in language processing is a bit overstated in the abstract and introduction and could be toned down. For example, Mani and Huettig (2016, LCN) discuss several arguments why prediction might be a tool used for comprehension, but might not be ‘central’ to comprehend language. Similarly, I do not agree with statements such as “[t]he ability to predict is essential for engaging in fluent dialogue” (p. 1; see for example Meyer et al., 2018 for recent experimental data). I do think though that it is an important task to further investigate and understand the mechanisms underlying predictive language processing and assessing its role in language acquisition, however, as the empirical evidence for prediction as ‘the central component’ of language comprehension is rather sparse, I would advise to formulate the introduction a bit more carefully in that regard.

Minor comment:

It seems as if references 1 and 2 as well as 10 and 11 were swapped in the text/references.

Author's Response to Decision Letter for (RSOS-180877.R0)

See Appendix A.

Decision letter (RSOS-180877.R1)

06-Nov-2018

Dear Ms Fazekas

On behalf of the Editor, I am pleased to inform you that your Manuscript RSOS-180877.R1 entitled "Do children learn from their mistakes? Evaluating error-based theories of language acquisition" has been accepted in principle for publication in Royal Society Open Science. The reviewers' and editors' comments are included at the end of this email.

You may now progress to Stage 2 and complete the study as approved. Before commencing data collection we ask that you:

- 1) Update the journal office as to the anticipated completion date of your study.
- 2) Register your approved protocol on the Open Science Framework (using this dedicated interface: <https://osf.io/rr>) or other recognised repository, either publicly or privately under embargo until submission of the Stage 2 manuscript. Please note that a time-stamped, independent registration of the protocol is mandatory under journal policy, and manuscripts that do not conform to this requirement cannot be considered at Stage 2. The protocol should be registered unchanged from its current approved state, with the time-stamp preceding implementation of the approved study design.

Following completion of your study, we invite you to resubmit your paper for peer review as a Stage 2 Registered Report. Please note that your manuscript can still be rejected for publication at Stage 2 if the Editors consider any of the following conditions to be met:

- The results were unable to test the authors' proposed hypotheses by failing to meet the approved outcome-neutral criteria.
- The authors altered the Introduction, rationale, or hypotheses, as approved in the Stage 1 submission.
- The authors failed to adhere closely to the registered experimental procedures. Please note that any deviations from the approved experimental procedures must be communicated to the editor immediately for approval, and prior to the completion of data collection. Failure to do so can result in revocation of in-principle acceptance and rejection at Stage 2 (see complete guidelines for further information).
- Any post-hoc (unregistered) analyses were either unjustified, insufficiently caveated, or overly dominant in shaping the authors' conclusions.
- The authors' conclusions were not justified given the data obtained.

We encourage you to read the complete guidelines for authors concerning Stage 2 submissions at <http://rsos.royalsocietypublishing.org/content/registered-reports>. Please especially note the requirements for data sharing, reporting the URL of the independently registered protocol, and that withdrawing your manuscript will result in publication of a Withdrawn Registration.

Please note that Royal Society Open Science will introduce article processing charges for all new submissions received from 1 January 2018. Registered Reports submitted and accepted after this date will ONLY be subject to a charge if they subsequently progress to and are accepted as Stage 2 Registered Reports. If your manuscript is submitted and accepted for publication after 1 January 2018 (i.e. as a full Stage 2 Registered Report), you will be asked to pay the article processing charge, unless you request a waiver and this is approved by Royal Society Publishing. You can find out more about the charges at <http://rsos.royalsocietypublishing.org/page/charges>. Should you have any queries, please contact opscience@royalsociety.org.

Once again, thank you for submitting your manuscript to Royal Society Open Science and we look forward to receiving your Stage 2 submission. If you have any questions at all, please do not hesitate to get in touch. We look forward to hearing from you shortly with the anticipated submission date for your stage two manuscript.

Kind regards,
Royal Society Open Science Editorial Office
Royal Society Open Science
opscience@royalsociety.org

on behalf of Professor Chris Chambers (Registered Reports Editor, Royal Society Open Science)
opscience@royalsociety.org

Author's Response to Decision Letter for (RSOS-180877.R1)

See Appendix B.

RSOS-180877.R2 (Revision)

Review form: Reviewer 2

Is the manuscript scientifically sound in its present form?

Yes

Are the interpretations and conclusions justified by the results?

No

Is the language acceptable?

Yes

Do you have any ethical concerns with this paper?

No

Have you any concerns about statistical analyses in this paper?

No

Recommendation?

Accept with minor revision

Comments to the Author(s)

See attached report (Appendix C).

Review form: Reviewer 3 (Arpitha Vasudev)

Is the manuscript scientifically sound in its present form?

Yes

Are the interpretations and conclusions justified by the results?

Yes

Is the language acceptable?

Yes

Do you have any ethical concerns with this paper?

Yes

Have you any concerns about statistical analyses in this paper?

Yes

Recommendation?

Accept with minor revision

Comments to the Author(s)

Our comments to authors have been attached (Appendix D).

Decision letter (RSOS-180877.R2)

Dear Dr Fazekas:

On behalf of the Editor, I am pleased to inform you that your Stage 2 Registered Report RSOS-180877.R2 entitled "Do children learn from their mistakes? A registered report evaluating error-based theories of language acquisition" has been deemed suitable for publication in Royal Society Open Science subject to minor revision in accordance with the referee suggestions. Please find the referees' comments at the end of this email.

The reviewers and Subject Editor have recommended publication, but also suggest some revisions to your manuscript. Therefore, I invite you to respond to the comments and revise your manuscript.

Please also ensure that all the below editorial sections are included where appropriate -- if any section is not applicable to your manuscript, please can we ask you to nevertheless include the heading, but explicitly state that the heading is inapplicable. An example of these sections is attached with this email.

- Ethics statement

- Data accessibility

If you wish to submit your supporting data or code to Dryad (<http://datadryad.org/>), or modify your current submission to dryad, please use the following link:
[http://datadryad.org/submit?journalID=RSOS&manu=\(Document not available\)](http://datadryad.org/submit?journalID=RSOS&manu=(Document not available))

- Competing interests

- Authors' contributions

- Acknowledgements

- Funding statement

Because the schedule for publication is very tight, it is a condition of publication that you submit the revised version of your manuscript within 30 days (i.e. by the 13-Aug-2020). If you do not think you will be able to meet this date please let me know immediately.

Please note that Royal Society Open Science will introduce article processing charges for all new submissions received from 1 January 2018. Registered Reports submitted and accepted after this date will ONLY be subject to a charge if they subsequently progress to and are accepted as Stage 2 Registered Reports. If your manuscript is submitted and accepted for publication after 1 January 2018 (i.e. as a full Stage 2 Registered Report), you will be asked to pay the article processing charge, unless you request a waiver and this is approved by Royal Society Publishing. You can find out more about the charges at <https://royalsocietypublishing.org/rsos/charges>. Should you have any queries, please contact openscience@royalsociety.org.

on behalf of Professor Chris Chambers
(Registered Reports Editor, Royal Society Open Science)
openscience@royalsociety.org

Associate Editor Comments to Author (Professor Chris Chambers):

Associate Editor: 1

Comments to the Author:

Thank you for your patience during this challenging time for reviewers. Three reviewers have now assessed the manuscript, and before summarising their assessments I want to explain how the review process was managed due to unusual circumstances.

You will recall that at Stage 1, one of the original reviewers (Kuppuraj Sengottuvel) sadly passed away. He signed his Stage 1 review and assessed the manuscript jointly with a junior colleague, Arpitha Vasudev. At Stage 2, Arpitha kindly agreed to return to assess the manuscript together with a senior colleague, Dr Prema Rao, who was Kuppu's mentor/PhD supervisor in India. Their joint review is signed and makes up one of the review reports ("Reviewer 3"). The other review is supplied by a new (anonymous) reviewer who was not part of the Stage 1 process (Reviewer 2), but has been very diligent in reviewing your submission according to the Stage 2 criteria.

As you will see, both reviewers are generally positive about your submission and the majority of comments and suggestions concern clarity of presentation (including some of the deviations from protocol) and the interpretation of the results in the Discussion. You will see that Reviewer 2 (in point 5) suggests that the original analysis be relegated to the supplementary materials. Although this is unusual for a Stage 2 RR, provided it is reported just as comprehensively in the supplementary information as now, I am open to this possibility if you also agree. I will leave you to consider this particular issue alongside the many other helpful recommendations from both sets of reviews. Note that addressing some of the comments will require minor changes to the Stage 1 part of the manuscript. I am happy for these minor changes to be made as they mainly involve clarification of deviations. The title can also be changed.

If you need any further specific guidance on responding to the comments feel free to contact me through the journal office (by emailing Open.Science@royalsociety.org)

Comments to Author:

Reviewer: 2

Comments to the Author(s)

See attached report

Reviewer: 3

Comments to the Author(s)

Our comments to authors have been attached below.

Author's Response to Decision Letter for (RSOS-180877.R2)

See Appendices E & F

Decision letter (RSOS-180877.R3)

Dear Ms Fazekas:

It is a pleasure to accept your manuscript entitled "Do children learn from their mistakes? A registered report evaluating error-based theories of language acquisition" in its current form for publication in Royal Society Open Science. Congratulations on a major piece of work and for successfully navigating a challenging review process.

Please note that 'a.a.jessop@liverpool.ac.uk' is not currently receiving messages from us - please can you either ask Dr Jessop to add emails from this address to their 'whitelist' or alternatively supply the editorial and production office with a different address for Dr Jessop?

Kind regards,
Andrew Dunn
Royal Society Open Science
openscience@royalsociety.org

on behalf of Professor Chris Chambers (Subject Editor)
openscience@royalsociety.org

Appendix A

Response to reviewers: RSOS-180877

Dear Chris,

First of all, we wanted to say how sorry we were to hear of Dr Sengottuvel's passing, and how much we valued his extremely helpful and constructive comments. We are also very grateful that you and the reviewers worked so hard to get this reviewed so quickly, especially over the summer season and after experiencing technical difficulties.

Below we detail how we have responded to the comments - responses in italics/blue below the comment. We have pasted in reviewer 2's comments (which were received separately) under those of reviewer 1, for ease of reading/reviewing.

Best wishes,

Judit

Associate Editor Comments to Author (Chris Chambers):

Associate Editor: 1

Comments to the Author:

At the outset, please accept my apologies for the slow handling of your submission, which does not meet the standards of promptness that we seek maintain at RSOS. The reason for the delay was that we experienced significant problems finding reviewers over the summer break, compounded by the fact that after quickly receiving one high-quality review, the other reviewer became non-responsive, and only after agreeing to the review on an extended deadline. The manuscript has therefore received one in-depth review, in this case provided jointly by two reviewers. Based on this review and my own assessment (including the revisions made during the previous editorial triage round) I have decided to proceed on this basis to avoid further delay.

Should your manuscript eventually receive IPA and be completed as planned, I may recruit an additional expert reviewer at Stage 2, although that reviewer will of course be bound to the Stage 2 criteria and will not be (re)assessing approved Stage 1 material.

The good news is that your manuscript was positively appraised by the reviewers who provided the assessment in Review 1. Please note carefully their recommendations concerning clarity of the Introduction, rationale for the hypotheses and current design, and provision of additional methodological detail. Provided you are able to respond thoroughly to the comments in Review 1, I would anticipate IPA to be forthcoming without requiring further in-depth review.

Reply: We have addressed all points, particularly with regard to clarity, rationale and methodological detail. We explain how we have responded to each comment in detail below.

We have submitted both a finalised version of the manuscript and an additional one where the changes are tracked.

Reviewer comments to Author:

Reviewer: 1

Comments to the Author(s)

To the authors

The authors examine whether surprising (less predictable) input leads to more lasting changes in language than predictable input using a novel approach called- prime surprisal. Despite my trivial concerns, I feel the review is adequate and motivates the research questions. The method is meticulously planned to address the research questions and the study warrants registered report route of investigation. While, I feel the design is already sufficient, I raise some points which the authors may want to clarify before the protocol is accepted in principle.

Introduction

Specific (must address)

1. Page 2-3: It is not clear to me what authors mean by ‘Second, rather than simply seeking to define children’s state of knowledge at different developmental stages, these models explain how children move from one knowledge state to another and hence provide a specific learning mechanism that can be tested experimentally’. An example (and a reference, if suitable) would make the reading easier.

Reply: We have included additional explanation and a reference on page 3 of the manuscript.

2. Page 4: Lines 38 onwards: Authors argue that previous findings may have been confounded with the process of integration. I am not sure, how their present design overcomes this flaw.

Reply: Rabagliati et al. (2015, Language, Cognition and Neuroscience) suggested that in the looking-while-listening paradigm (which is typically used to target predictions in childhood) instead of predicting the next word themselves, children might simply be choosing the most plausible sentence ending from those that are pictured. As our paradigm does not involve pictures of potential predictions the responses cannot be guided by the visual input.

We have clarified this on page 5 of the manuscript.

3. Page 7: Lines 33:35. Theories (and authors) argue that the error signal leads to learning. For instance, when the child hears a violating (DOD with ‘bring’) phrase, the child generates an error signal. This assumes that the child is aware of what the usual occurrence pattern is, that is, DOD with ‘give’. So, for this design to work, child must have already acquired/mastered this knowledge (the end product of learning?)

If authors indeed agree with my above argument, I would like to question the error based theories' significance in language acquisition research. That is, for it to work you need (a well developed ?) language in place. Authors are requested to clarify this (in paper if it is sensible or to me if my argument is baseless). For me this is a serious question, because, if authors agree with me, the present study could only answer 'whether error based priming surprisal effect exist / demonstrable' but NOT to the bigger question of whether error based learning is crucial for (language) learning.

Or

This is where the comparison with adults (fully developed system) and children (malleable system) is justified? [see my point 3 & 4 in Method]

Reply: According to error-based theories, this mechanism is crucial from the start of the language acquisition process. Children make predictions from a very early age, but since these predictions are based on very little (or no) accumulated knowledge they are often incorrect. For instance a child with very little knowledge could in principle predict that the next word after "gave" is null (no word): it would be wrong and thus this would lead to learning. Once children gain more linguistic experience their predictions get more accurate and at the same time their linguistic knowledge becomes less malleable. So as the reviewers point out children at a young age (5-6) would already have some verb-structure preferences learned from the input via error-based learning but (as they are based on less input) they are more malleable than adults' biases. Children's weaker representations should lead to more learning (stronger priming effects and post-test shift) while the adult's more developed system would be less sensitive to the error signals produced to unexpected sentences leading to smaller effects in our study.

We have included this explanation on page 7 in the manuscript.

General

1. Is this 'error based learning' mechanism unique language learning mechanism? Or is it proposed as a general learning mechanism that is present across animal species?

Reply: While error-based learning does not seem to be specific to language in humans, we are not aware of any work on error-based learning in animals. We have included a note and some references on error-based learning in humans in domains other than language on page 2 in the manuscript.

Method

Specific

1. Is there any reason why authors want the experimenter to describe (the primes) and not use a pre-recorded description? A recorded material brings uniformity between trials and participants (The voice of the experimenter is prone to mood/emotional changes). However, I

do believe that authors have a reason to opt for this approach. It may be worth explaining in the text.

Reply: The bingo paradigm used in this study has been very successful in engaging children in similar experiments in the past and it also led to the production of a high number of target structures (e.g. Peter et al. 2015, JML; Rowland et al. 2012, Cognition). In this paradigm the priming study is embedded in a bingo game which relies on back-and-forth communication between the participants and the experimenter. The interactive and playful nature of the study would be hard to maintain with pre-recorded instructions.

2. Page 10: Line 52, Should this '(i.e. DODs for participants in group A)' read as '(i.e., PD's for participants in group A)'? Or did I get it wrong. Please clarify!

Reply: It should indeed read as (i.e. PDs for participants in group A), we have corrected this.

3. Must justify more strongly the inclusion of adults in the design? Is it a reference sample to validate the design?.Page 13: Line 26: 'These age groups have shown sensitivity to verb-bias manipulations both in the target verb and in the prime verb (prime surprisal) conditions in a priming study involving dative structures'. Then why have adults? The results may be easier to interpret if we EXCLUDE adults from the analysis.

Reply: The different predictions of error-based learning theories (Discussed at Specific comments point 3) for children and adults are the main reason to include adults in our design. This means we can examine not only whether error-based learning is active in childhood but also if it stays active in adulthood, as well as whether learning effects are stronger in children as compared to adults.

4. Page 20: Lines 37-40 'but, in order to explore the group-specific patterns in more detail, we will also carry out analyses on the data from the two age groups separately.' Why is this necessary? Do the authors have differential predictions for these groups that are theoretically motivated? Without that stated, I assume that the authors have little interest in adult's group. In which case, I suggest the authors to drop the adults from the plan and focus on children. Or if they do feel adults are needed they must justify the inclusion in the manuscript. If their justification is weak, I would like to only have a small adult sample (preliminary, exploratory) and see if they see the effects and then take it to children.

Reply: We have stated the different prediction for children and adults more clearly in the manuscript on Pages 7 and 17.

Note: I am not insisting on the changes in 3 & 4. I feel the authors have some assumptions that need explicitly stating in the manuscript. I leave it to authors convenience to work it out or stay with the same design if they could justify having adults.

5. Page 15, Design section. I would like to see the predictions stated right in this section for ease of following.

Reply: We have stated the predictions after the Design section in section 2.3. Predictions on page 16 of the manuscript.

6. Given the argument that error based learning might underlie language acquisition, would it be nice to have a language measure and see if participant's ability to generate error signal/ or learn from the surprisal effect predict their language proficiency?

Reply: We will include the Sentence Imitation task from the Early Repetition Battery to measure morphosyntactic abilities. See Page 20 of the manuscript.

7. Page 18: Line 26: Check the bracket usage

Reply: We have corrected the bracket usage.

8. Since the aim is to examine whether the learning that results is 'lasting', shouldn't there be another post-test phase after a delay? Could the present post-test be broken in to two? (immediate and delayed).

Reply: We agree that it would be very useful to have a delayed post-test. However an extra testing session would make both the recruitment and the testing itself much more challenging, especially for the child participants. This would potentially threaten the completion of the project in the given time-frame.

Kuppuraj Sengottuvel (Newton Fellow and Western research Fellow) kuppuslp@gmail.com

&

Arpitha Vasudev (Post graduate in Speech Language Pathology) arpithakullal@gmail.com

Reviewer 2

Fazekas et al.: Do children learn from their mistakes? Evaluating error-based theories of language acquisition

Review

The registered report presents an interesting idea for a study that is very likely to advance our understanding of the mechanisms underlying language acquisition. The study design is clear; the method seems appropriate and is well motivated by previous research. The number of participants is sufficient and supported by power calculations. The hypotheses are clearly formulated and plausible.

My suggestions for the authors concern two issues: the statistical analysis and the framing. In the proposal, the authors write that they plan to conduct mixed effects analyses. In fact, they intend to "fit maximal models and simplify the random effects structure until the model converges" (p. 20). This does not seem to be the right practice. Note that model non-convergence is not the same as a given predictor not having an effect. The paper by Barr et al. (2013, JML) suggests to simplify the model structure when predictors do not show an effect and subsequently compare the slimmer model with the previous one to check whether the adjusted model does a better job at explaining the data. I gather that the authors are aware that using mixed-effect models on categorical data often leads to model non-convergence, an observation also reported by Barr and colleagues: "[...] practical experience with fitting

LMEMs to datasets with categorical response variables, in particular with mixed logit models, suggests more difficulty in getting maximal models to converge” leading them to conclude that “although our arguments and recommendations [...] still apply in principle, they might need to be modified for non-continuous cases” (see Baayen et al., 2017, for further discussion). In light of the lack of a clear standard for mixed-effects analyses on categorical data I urge the authors to complement their analyses using other techniques, i.e. magnitude estimation (Cumming, 2014) or Bayesian approaches (Bolker et al., 2009, TiEE, for further discussion).

Reply: We are very grateful for the reviewer’s comments on our analysis strategy but feel that there has been a slight misunderstanding of our approach. We would like to clarify the logic and basis of our proposed strategy.

Our approach follows the analyses of other priming studies with categorical outcome measures (e.g. Peter et al., 2015; JML, Branigan & Messenger, 2016, Cognition). We are planning to fit generalised linear mixed-effects models with the maximal random effects structure (Baayen et al., 2008; Barr et al., 2013) using the lme4 package (Bates et al., 2015). As the reviewer has stated, these models often fail to converge depending on the complexity of the random effects structure and whether it can be supported by the available data (Barr et al., 2013; Bates, Keigl, Vasishth & Baayen, 2015). This non-convergence means that the model has been unable to arrive at “best estimates” for the included parameters. Under such circumstances, the critical statistics produced by this model – namely, the beta estimates and their standard error – are unreliable and therefore cannot be trusted or be used to make meaningful generalisations. Therefore, it is necessary to use a model structure that does converge and can be supported by the available data. To increase the likelihood of being able to use a maximal random effects structure, we have run power simulations to ensure that a large enough sample size is recruited to properly support this model. However, it may still be necessary to reduce the complexity of the random effects structure. There are different strategies available for such model simplification. One option is to use principle components analysis, which aims to identify the greatest number of dimensions that can be included in the model (Bates, Keigl, Vasishth & Baayen, 2015). However, we have opted for the simpler approach of gradually reducing the random effects structure, starting from the highest order terms (interactions). In both of these approaches, it is necessary to remove some of the random effects in order to obtain reliable estimates, although the exact terms that are removed may differ. We would like to emphasise that we will not be removing any of the hypothesised predictors (i.e., the fixed effects). Our simplification procedure will only affect the random effects structure of the model. We have clarified this in the manuscript on Page 21.

Additionally, as suggested by the reviewer, we will provide Bayesian analyses. Specifically, the Bayes factor will be calculated using the uniform distribution to provide estimates for the degrees of confidence in the results. This will be particularly useful should the hypothesized effects produce null results, as it will allow us to assess whether the data provides evidence for a true null effect or whether the study lacks sensitivity to detect the effects (see Dienes, 2014). We will also combine this with a post-hoc power analysis and bootstrapped confidence intervals for the model estimates to provide a comprehensive assessment of the reliability of the results.

Moreover, to my taste, the description of the role/importance of prediction in language processing is a bit overstated in the abstract and introduction and could be toned down. For

example, Mani and Huettig (2016, LCN) discuss several arguments why prediction might be a tool used for comprehension, but might not be ‘central’ to comprehend language. Similarly, I do not agree with statements such as “[t]he ability to predict is essential for engaging in fluent dialogue” (p. 1; see for example Meyer et al., 2018 for recent experimental data). I do think though that it is an important task to further investigate and understand the mechanisms underlying predictive language processing and assessing its role in language acquisition, however, as the empirical evidence for prediction as ‘the central component’ of language comprehension is rather sparse, I would advise to formulate the introduction a bit more carefully in that regard.

Reply: We have modified the abstract and introduction in order to accommodate views in which predictions are less central in human communication.

Minor comment:

It seems as if references 1 and 2 as well as 10 and 11 were swapped in the text/references.

Reply: We have corrected the references.

Appendix B

Judit Fazekas
Language Development Department
Max Planck Institute for Psycholinguistics
Wundtlaan 1
6525 XD Nijmegen
The Netherlands

Email: Judit.Fazekas@mpi.nl
Tel: +31 24 3521371

June, 2020

Dear Editors,

We are pleased to resubmit our manuscript, “Do children learn from their mistakes? A registered report evaluating error-based theories of language acquisition” by Judit Fazekas, Andrew Jessop, Julian Pine and Caroline Rowland as a Stage 2 registered report. We apologize for the delay to resubmission, which was due to the scheduling needs of participating schools as well as the more recent disruptions.

The manuscript contains the URL for archived study data, digital materials/code and the laboratory log on page 61 and the URL for the approved Stage 1 protocol on the Open Science Framework on page 11. We can confirm that no data was collected for the current study prior to the date of the IPA.

Please note that we had some minor deviations from our pre-registered protocol. First, we introduced a new measure of syntactic knowledge in case participants showed a ceiling performance on our original test (this change was agreed on by the editorial office on 30/03/2019 and is noted in the manuscript on page 19). Second, we mistakenly pre-registered a binary as opposed to a continuous predictor in one of our analyses in our stage 1 report. We have corrected this and discussed the modification on page 36.

We also made three minor adjustments to the original manuscript (that did not affect how the study was carried out) to ease readability or correct previous mistakes. First, we added “registered report” to the title of the manuscript. Second, we made a black and white version of the table on page 12 to ensure legibility in all formats. Third, we detected some unintentional repetitions in the example sentence list attached to the Stage 1 Registered Report. We have now corrected this and made a note in the manuscript (on page 54). Please let us know whether there are any issues with these changes.

We look forward to your and the reviewers’ comments on the manuscript.

Yours sincerely

Judit Fazekas
Max Planck Institute for Psycholinguistics
(for all co-authors)

Appendix C

Fazekis: Do children learn from their mistakes?

1. The authors are to be congratulated on completing a complex piece of work. The pre-registration, open materials and data, and immaculately presented scripts give confidence in the findings and set a very high bar for other researchers.
2. I felt the authors did a good job of explaining the logic of the paper and steering the reader through a complex set of analyses and what they meant. The design was clever and the materials very well-constructed. Nevertheless it was not always easy to follow the thread, some analyses seemed overcomplicated, and I had major difficulties the interpretation of results (explained more below).
3. The methods follow the pre-registration except in some details that are discussed later in the report; the changes all have sensible rationale, namely that the original approach would give ceiling effects.
4. I appreciate it is not possible to modify the sections that were pre-registered, but I did find it hard to understand why the adult group was included, given that the focus was on how children learn language. Predictions are made that children will show larger effects than adults, but the question really is why would adults show any effects at all, given that they are deemed to be competent language users. Insofar as they do show the predicted effects, doesn't that rather weaken the interpretation of the effect as indicating language learning? This, indeed, relates to my key criticism of the Discussion (see below)
5. There are some departures from the specified analysis, but they are well motivated. In particular, the plan had been to measure change from a baseline period in terms of rates of production of sentence frame types, but some participants were already producing 100% at baseline. This makes the use of a change score unsuitable for addressing the research question, and the authors explain this and then present an alternative analysis omitting those who were at ceiling. My preference would be to relegate the original analysis to supplementary materials, since it is not appropriate – the authors can then show it was done, but I think it is important that the paper, already very dense with analyses, does not get bogged down with unnecessary detail. At the end of the day, the point of preregistration is to keep everyone honest, not to make papers indigestible.

6. Main point of criticism

I found the interpretation of findings in the Discussion was problematic. It is possible that I have misinterpreted some aspect of the design, as it is complex and outside my current research area. But I struggled with the idea that this study is telling us about language learning.

The rationale (line 42) is that 'Every time they make an incorrect prediction, linguistic representations change, which, in children, moves them a step closer to the adult state'. The idea of the study is to assess 'whether less predictable input leads to more lasting change than more predictable input'. Furthermore 'the key prediction of this account is that incorrect predictions lead to learning. To test this, we need to demonstrate that prime surprisal leads to lasting cumulative language change as well.' (p 7)

Yet what was found was that a proportion of both children and adults showed an increase in production of a primed sentence construction when the primed sentence had been presented with a verb that was not usually used with that construction.

The effect was statistically significant but numerically small; assuming I have understood Figure 2 correctly, around 1/3 children showed an effect in the opposite direction to prediction.

A manipulation check (phase 2) failed (section 3.2.3): "neither Hypothesis 1a (replication of prime surprisal effects) nor Hypothesis 1b (larger prime surprisal effects in the child than in the adult group) were supported by the current dataset." Surely this is of concern because it undermines the mechanism that is postulated to explain the gains in phase 3? There is some discussion of this on p. 44.

In a later phase there was a further test where the same verbs were used as had been used in priming. 'If there is verb-specific error-based learning, we expect an enhanced shift towards the dative structure the verb previously appeared with when the structure did not match the verb's bias.' (p. 10). But in fact, the use of the primed construction dropped away. Thanks to the excellent documentation of scripts and data I was able to look at the data in a slightly different way, and this confirmed that at this 2nd post-test, the effect seen at 1st post-test bounced back to be, if anything, more extreme than the baseline. I may have misunderstood this, but having just looked at means in an initial data check, this did seem inconsistent with expectation.

phase	bgro	agroup	dod mean	sd
	up			
Baseline	D	adult	0.635	0.482
Post-test	D	adult	0.668	0.472
Second post-test	D	adult	0.598	0.491
Baseline	P	adult	0.68	0.467
Post-test	P	adult	0.628	0.484
Second post-test	P	adult	0.74	0.44
Baseline	D	child	0.291	0.455
Post-test	D	child	0.364	0.482
Second post-test	D	child	0.177	0.383
Baseline	P	child	0.355	0.479
Post-test	P	child	0.306	0.462
Second post-test	P	child	0.371	0.484

A related question is whether the temporary shift in preference for one construction over another at first post-test can be regarded as 'moving a step closer to the adult state'. It's unclear how this could be given that we are already told that 'Children of this age consistently produce both PD and DOD structures (with an average DOD production of approximately 30%) in corpus-based studies.' So in what sense have they not already attained 'the adult state'. And the adults are presumably already in 'the adult state'.

So it is not clear that these results are relevant for understanding language *learning*. What has been shown is that priming, and predictability of structural variants, can induce a

temporary shift in preference for one version of a construction than the other, in children who already are familiar with both forms.

The study can be regarded as demonstrating, therefore, that children have some sensitivity to the 'surprisingness' of incoming sentences, but I do not think that this has been shown to influence language learning. It looks more like a kind of delayed priming of preference for one form over another – and given the results from the second post-test, this does seem temporary. This result is itself of interest.

More minor points

7. I find the title non-optimal. I had assumed from the title that there would be some analysis of children's mastery of grammar in relation to their making grammatical errors. This kind of thing is discussed in relation to prior literature on p 3. But this study is not about children making linguistic errors: rather, it's about them learning from hearing familiar constructions in surprising sentence contexts. The 'error' that is referred to is assumed: a mismatch between prediction and what is heard. This is pretty confusing. So I think a more accurate title would be 'Do children learn more from unpredictable input?' or 'Do children learn more when input does not match predictions?'. Though given my reservations about what has been shown, maybe avoid mentioning 'learning' too, e.g. – 'Are children sensitive to predictability of language input?'

8. One change from pre-registration is that The Test for Reception of Grammar was substituted as a covariate to assess language ability instead of the Sentence Imitation task, because the latter gave ceiling effects. That is adequate justification. However, I would ask that one clarification be made that would involve modifying the original pre-registration. It is unclear which version of TROG was used (the test is not referenced; is this TROG or TROG-2) and how it was scored. The usual scoring is in number of blocks correct, with a range from 0 to 20. This gives a raw score which increases with age between 5 and 6 years. This can be converted in to an age-scaled score or percentile. We need to know which of these was used, especially as the presented data appear to be on a different scale. The Sentence Imitation task also should be referenced.

9. Table 7: it would help if the labels for the phases were added, to match Table 1.

Appendix D

General remarks

The study is carried out with a well-planned design. Justification for the choice of design is well supported by studies (Page 9/62: Para 1 and Para 2). Sampling based on power calculation with cushion for attrition is also appreciable (Page 15/62). Variables under study are clearly indicated (Page 16/62). The data has been viewed at micro-level of performance followed by systematic analysis. While both the Hypothesis 1a (replication of prime surprisal effects) and Hypothesis 1b (larger prime surprisal effects in the child than in the adult group) were not supported by the results (Page 34/62: Line: 33-52), sincere acknowledgement of findings derived contrary to the hypotheses needs a mention. Consequently, the authors devised an approach to tease apart data at micro-levels to carry out further extensive analyses for better know-how of the complexities of the construct of error-based theories of language acquisition. This reflects the authors' depth of understanding about research process and significance of analysis.

Please comment explicitly on each of the following points in your comments to the authors:

Specific remarks

1. Whether the data are able to test the authors' proposed hypotheses by passing the approved outcome-neutral criteria (such as absence of floor and ceiling effects or success of positive controls):

The authors report that the analysis of the data for examining the main objective (H2b), the main effect of the shift did not reach significance in both the groups and the lack of significant effects in children was attributed by the authors to the ceiling effects in this group (Page 44/62; line: 48-50). To control for ceiling effect, the data of 14 children that showed ceiling effects were eliminated in exploratory analysis (but not in main analysis) that is quite acceptable. But, such reduction in the sample size (from 72 to 58) could have potentially reduced the sampling power that has not been calibrated/ calculated and addressed in the report. **Authors may clarify this in the revised report.**

Despite the suggestion made for Stage 1 report (Reviewer -1), there was no attempt to carry out language assessment before the selection of participants for Group 1 (children group). Instead, the SIT and TROG were administered at a later stage (after the experimental task – BINGO game completion). Also, since the scores on SIT reached ceiling at this stage, the test scores were not considered for analysis (Page 19; Line 51-54). To the best of our knowledge, the Sentence Imitation Test from Early Repetition Battery Test (SIT; Seeff-Gabriel, Chiat, and Roy, 2008), for example, contains a comprehensive range of simple sentence structures sampling a comprehensive range of **function words** in English. This provides information about the aspects of sentences that children find difficult, which is very useful in planning therapy. Therefore, the rationale for the choice of this test as a part of language assessment is unclear in the report.

(Source:

https://www.researchgate.net/publication/259390258_The_potential_of_sentence_imitation_tasks_for_assessment_of_language_abilities_in_sequential_bilingual_In_Mueller_Gathercole_VC_Ed_Bilinguals_and_assessment_State_of_the_art_guide_to_issues_and_solution

The potential of sentence imitation tasks for assessment of language abilities in sequential bilingual. In: Mueller Gathercole, V.C. (Ed.). Bilinguals and assessment: State of the art guide to issues and solutions from around the world).

In view of the above, the additional time and efforts of the authors besides taxing children with longer duration of testing time could have been avoided had the authors taken care to choose the test with justification (say, only TROG). Also, in our opinion, the floor and ceiling effects could have been better controlled to address the main objective of the study with adequate sampling power without resorting to exploratory analysis stages. Our remarks also receive support from authors' statement in Page 34/62: Line: 16-22: 'Four-way interaction suggests that children who are younger and have lower TROG scores are more likely to show a sensitivity to the prime surprisal manipulation (the interaction of prime structure and prime bias-match)'. The task is based on priming phenomenon that facilitates learning (syntax units that are weak) or accelerating (if the units are in the process of development) similar to how it works for clinical population. This endorses our remarks (and suggestion by Reviewer 1 for Stage 1 report) that baseline for language should have been established before sampling, particularly for the child participants.

The authors may clarify the selection of tests and the reason for not considering language assessment as suggested by Reviewer 1 for Stage 1 report.

2. Whether the Introduction, rationale and stated hypotheses are the same as the approved Stage 1 submission.

Yes. Introduction, rationale and stated hypotheses are the same as the approved Stage 1 submission. However, in response to one of the previous reviewer's comments, the authors have made slight modification of a statement in method section (Page 11/62; line: 11-18) 'Here, participants described target video animations depicting transitive actions in a similar way to the baseline phase, but the experimenter preceded these participant descriptions by describing prime animations using either DOD or PD structure'. Yet, this modified statement fails to address the query by Reviewer -1) stated as 'Is there any reason why they want the experimenter to describe (the primes) and not use a pre-recorded description? A recorded material brings uniformity between trials and participants (The voice of the experimenter is prone to mood/emotional changes). However, we do believe that authors have a reason to opt for this approach. It may be worth explaining in the text'.

The authors may state the reasons in Stage-2 Report.

In addition, we recommend that the instructions given to the participants before the task should be stated in the report for more clarity and replication of the study, if any, in the future.

3. Whether the authors adhered precisely to the registered experimental procedures

Yes, authors have substantially adhered to all the registered experimental procedures. However, in statistical analysis, authors have deviated from pre-registered analysis (Hypotheses 3a) for which justification is given in result section.

4. Where applicable, whether any unregistered exploratory statistical analyses are justified, methodologically sound, and informative.

- a) Yes, given the initial analyses of the results being not supportive to the objectives of the study, the decision made by the authors to plan for exploratory analysis in each subsections of results detailed as under:

- i. By teasing apart data [in hypothesis 2a, 21 adults and 14 children that showed a ceiling performance in the pre-test phase],
- ii. By changing choice of design and re-analysing the data (in hypothesis 1a, by adding base line DOD pre-test score) and,
- iii. By altering the pre-registered analysis design (in hypothesis 3a, interchanging independent variable with those appropriate- prime structure, prime bias match and verb bias match- to test H3a).

The above modifications are considered as justified, methodologically sound, and informative.

- b)* We feel that the statistical reasoning and justification for dropping Bayesian analysis in exploratory analysis could be made more explicit for the readers instead of just stating that it was not carried out (Page 34/62; line:15, Page 36/62; line: 56-58 and Page 41/62; line 30-33).
- c)* In view of the complexity of the analyses, the authors may indicate the process of analysis across the four phases through a flowchart before describing the results of analyses.

5. Whether the authors' conclusions are justified given the data

Fair conclusion is drawn based on the data and analyses. Limitations are also specified but appears to be restricted to only three limitations. While considerable attempt is made by the authors to provide an extensive review, the reviewed articles suggest certain other limitations as detailed below.

Additional remarks

- a) Authors have inconsistently used the terms, long lasting effects, lasting learning effects, and lasting priming effects in the manuscript. We suggest that an operational definition may be given for these terms if they wish to use all the three terms or decide on consistent use of any one term.
- b) Page 6/62; line: 29-33 'Long term' has to be explicitly specified. Is it with reference to hours, days or months?
- c) Page 6/62; line- 32-39; The authors mention that 'This method not only provides us with information about the immediate and longer-term outcome of correct and incorrect predictions, but also overcomes the problems inherent in using the looking while-

listening paradigm, as it does not involve pictures of more or less predictable sentence endings, and so the responses cannot be guided by visual input’.

But the primary task in the study is description of animated cartoon characters presented through videos. The participants are instructed to describe it with stem sentence. It does involve visual input and response is guided via visual input after listening to the prime-sentence produced by the investigator. Based on the descriptions given by the authors, we feel the paradigm in the present study is closer to visual (looking) while listening task- there is a need for clarity here.

Clarification may be offered in the report.

- d) ‘There were no error??? verb-based learning effects in general but there was only reliable evidence for immediate prime surprisal effects in the adult, but not in the child group.’ (Page 44/62; line 54-58). Does it suggest that language adequacy probably plays a role in prime-surprisal? As there was no baseline established for language level even in Stage 2 study, the results could have been confounded by this factor. (We invite input to our understanding if this is incorrect).
- e) Page 25/62: line 17-24: The presence of immediate effects and the absence of lasting effects would show evidence of prediction but no evidence of learning via prediction. Given the complexity of the design and analysis, we suggest authors to supplement such descriptions with examples, wherever possible to avoid wordy narratives in the text.
- f) The assumption made by authors with regard to priming, prediction and learning as being aligned on a continuum needs supporting definitions or descriptions. Footnotes or endnotes may be given (if RSOS policy permits). This suggestion is made considering the debate in academia and research regarding the above three constructs. See...
- Dell, G. S., & Ferreira, V. S. (2016). Thirty years of structural priming: An introduction to the special issue. *Journal of Memory and Language, 91*, 1-4.
 - Rowland, C. F., Chang, F., Ambridge, B., Pine, J. M., & Lieven, E. V. (2012). The development of abstract syntax: Evidence from structural priming and the lexical boost. *Cognition, 125*(1), 49-63.
 - Huettig, F., & Mani, N. (2016). Is prediction necessary to understand language? Probably not. *Language, Cognition and Neuroscience, 31*(1), 19-31.

- Tooley, K. M., & Traxler, M. J. (2018). Implicit learning of structure occurs in parallel with lexically-mediated syntactic priming effects in sentence comprehension. *Journal of memory and language*, 98, 59-76.
 - Jones, T., & Farrell, S. (2018). Does syntax bias serial order reconstruction of verbal short-term memory?. *Journal of Memory and Language*, 100, 98-122.
- g) Authors claim to test long lasting/long term learning in the study. However, this study did not employ any offline testing to check lasting effect rather the procedure in second post-test was a part of bingo game not much different from the pre-test.
- h) Syntax learning is not merely a language activity but a cognitive-linguistic activity (Kurland, 2011). No attempt has been made (at least for children group in which the authors have shown large interest to analyse data) by the authors to examine vigilance and attention, given the fact that the entire task lasted for 45 + minutes in a single session.

Minor concerns to be addressed:

1. Page 6/62: Line 22: 'Visual world paradigm' – typo error?
To be changed as 'visual word paradigm'
2. Table 1: Heading
We would like to suggest authors to shorten the heading by keeping the description as Note: below Table or as a small paragraph.
3. Page 14/62; line: 16, Second **Pot** test Typo error? To be changed to-Second **post**-test?
4. Page 19/62; line: 31, the rationale for using semi randomised approach may be made clearer (add justification).
5. Page 19; line: 35-40 what is the time gap b/w pre and post and second post-test?
It's important to note the break or the gap durations.
6. Page 23/62; line: 44- 60? Component 'c' is missing?
 - a) bias group
 - b) pre-test
 - c) ???
 - d) age group
7. Sample test items are provided in method. Therefore, the appendix with the elaborate list may be planned as electronic material it is suggested the complete list may be given as electronic.

Appendix E

General remarks

The study is carried out with a well-planned design. Justification for the choice of design is well supported by studies (Page 9/62: Para 1 and Para 2). Sampling based on power calculation with cushion for attrition is also appreciable (Page 15/62). Variables under study are clearly indicated (Page 16/62). The data has been viewed at micro-level of performance followed by systematic analysis. While both the Hypothesis 1a (replication of prime surprisal effects) and Hypothesis 1b (larger prime surprisal effects in the child than in the adult group) were not supported by the results (Page 34/62: Line: 33-52), sincere acknowledgement of findings derived contrary to the hypotheses needs a mention. Consequently, the authors devised an approach to tease apart data at micro-levels to carry out further extensive analyses for better know-how of the complexities of the construct of error-based theories of language acquisition. This reflects the authors' depth of understanding about research process and significance of analysis.

Please comment explicitly on each of the following points in your comments to the authors:

Specific remarks

1. Whether the data are able to test the authors' proposed hypotheses by passing the approved outcome-neutral criteria (such as absence of floor and ceiling effects or success of positive controls):

The authors report that the analysis of the data for examining the main objective (H2b), the main effect of the shift did not reach significance in both the groups and the lack of significant effects in children was attributed by the authors to the ceiling effects in this group (Page 44/62; line: 48-50). To control for ceiling effect, the data of 14 children that showed ceiling effects were eliminated in exploratory analysis (but not in main analysis) that is quite acceptable. But, such reduction in the sample size (from 72 to 58) could have potentially reduced the sampling power that has not been calibrated/ calculated and addressed in the report. **Authors may clarify this in the revised report.**

Reply: We have clarified this in the report on page 24.

Despite the suggestion made for Stage 1 report (Reviewer -1), there was no attempt to carry out language assessment before the selection of participants for Group 1 (children group). Instead, the SIT and TROG were administered at a later stage (after the experimental task – BINGO game completion). Also, since the scores on SIT reached ceiling at this stage, the test scores were not considered for analysis (Page 19; Line 51-54). To the best of our knowledge, the Sentence Imitation Test from Early Repetition Battery Test (SIT; Seeff-Gabriel, Chiat, and Roy, 2008), for example, contains a comprehensive range of simple sentence structures sampling a comprehensive range of **function words** in English. This provides information about the aspects of sentences that children find difficult, which is very useful in planning therapy. Therefore, the rationale for the choice of this test as a part of language assessment is unclear in the report.

(Source:

https://www.researchgate.net/publication/259390258_The_potential_of_sentence_imitation_tasks_for_assessment_of_language_abilities_in_sequential_bilingual_In_Mueller_Gathercole_VC_Ed_Bilinguals_and_assessment_State_of_the_art_guide_to_issues_and_solution

The potential of sentence imitation tasks for assessment of language abilities in sequential bilingual. In: Mueller Gathercole, V.C. (Ed.). Bilinguals and assessment: State of the art guide to issues and solutions from around the world).

In view of the above, the additional time and efforts of the authors besides taxing children with longer duration of testing time could have been avoided had the authors taken care to choose the test with justification (say, only TROG). Also, in our opinion, the floor and ceiling effects could have been better controlled to address the main objective of the study with adequate sampling power without resorting to exploratory analysis stages. Our remarks also receive support from authors' statement in Page 34/62: Line: 16-22: 'Four-way interaction suggests that children who are younger and have lower TROG scores are more likely to show a sensitivity to the prime surprisal manipulation (the interaction of prime structure and prime bias-match)'. The task is based on priming phenomenon that facilitates learning (syntax units that are weak) or accelerating (if the units are in the process of development) similar to how it works for clinical population. This endorses our remarks (and suggestion by Reviewer 1 for Stage 1 report) that baseline for language should have been established before sampling, particularly for the child participants.

The authors may clarify the selection of tests and the reason for not considering language assessment as suggested by Reviewer 1 for Stage 1 report.

Reply: We added the additional language assessment (originally the SIT) based on reviewer 1's feedback at stage 1 review. However, we thought that the reason for this was to add exploratory power

to the outcome of the main study, and not to select participants. The SIT was originally selected to target the individual differences in children's morphosyntactic abilities. We then decided to use the TROG instead, as other research group members found ceiling effects with the SIT in a similar participant group. At this stage we consulted the editor to change the SIT to the TROG, and he suggested that we carry out both tests (as administering the SIT only takes approximately 5 minutes) and that we only use the TROG instead of the SIT if we found ceiling effects with the TROG. We discuss this in the report on pages 17-18.

2 Whether the Introduction, rationale and stated hypotheses are the same as the approved Stage 1 submission.

Yes. Introduction, rationale and stated hypotheses are the same as the approved Stage 1 submission. However, in response to one of the previous reviewer's comments, the authors have made slight modification of a statement in method section (Page 11/62; line: 11-18) 'Here, participants described target video animations depicting transitive actions in a similar way to the baseline phase, but the experimenter preceded these participant descriptions by describing prime animations using either DOD or PD structure'. Yet, this modified statement fails to address the query by Reviewer -1) stated as 'Is there any reason why they want the experimenter to describe (the primes) and not use a pre-recorded description? A recorded material brings uniformity between trials and participants (The voice of the experimenter is prone to mood/emotional changes). However, we do believe that authors have a reason to opt for this approach. It may be worth explaining in the text'.

The authors may state the reasons in Stage-2 Report.

In addition, we recommend that the instructions given to the participants before the task should be stated in the report for more clarity and replication of the study, if any, in the future.

Reply: We have included our rationale for not using pre-recorded materials in our response to reviewers in the previous review stage and have now added this to the manuscript as a footnote together with the instructions (on page 18).

3 Whether the authors adhered precisely to the registered experimental procedures

Yes, authors have substantially adhered to all the registered experimental procedures. However, in statistical analysis, authors have deviated from pre-registered analysis (Hypotheses 3a) for which justification is given in result section.

4 Where applicable, whether any unregistered exploratory statistical analyses are

justified, methodologically sound, and informative.

- a) Yes, given the initial analyses of the results being not supportive to the objectives of the study, the decision made by the authors to plan for exploratory analysis in each subsections of results detailed as By changing choice of design and re-analysing the data (in hypothesis 1a, by adding base line DOD pre-test score) and,
 - i. altering the pre-registered analysis design (in hypothesis 3a, interchanging independent variable with those appropriate- prime structure, prime bias match and verb bias match- to test H3a].

The above modifications are considered as justified, methodologically sound, and informative.

- b) We feel that the statistical reasoning and justification for dropping Bayesian analysis in exploratory analysis could be made more explicit for the readers instead of just stating that it was not carried out (Page 34/62; line:15, Page 36/62; line: 56-58 and Page 41/62; line 30-33).

Reply: We included a Bayesian analysis for every frequentist analysis. However, we have dropped some of the predictors from the Bayesian analyses, when the full models had convergence issues. We have included a detailed description of the Bayesian model selection on page 20.

- c) In view of the complexity of the analyses, the authors may indicate the process of analysis across the four phases through a flowchart before describing the results of analyses.

Reply: We included an analyses flowchart in the manuscript.

5. Whether the authors' conclusions are justified given the data

Fair conclusion is drawn based on the data and analyses. Limitations are also specified but appears to be restricted to only three limitations. While considerable attempt is made by the authors to provide an extensive review, the reviewed articles suggest certain other limitations as detailed below.

Additional remarks

- a) Authors have inconsistently used the terms, long lasting effects, lasting learning effects, and lasting priming effects in the manuscript. We suggest that an operational definition may be given for these terms if they wish to use all the three terms or decide on

consistent use of any one term.

Reply: The manuscript does not use “long lasting” with reference to the current study (as based on the current results we cannot determine the exact length of time these effects persist). The terms “lasting learning” or “lasting language change” are used to contrast with the immediate priming effects our test phase targeted. We have added a footnote on page 5 discussing this.

- b) Page 6/62; line: 29-33 'Long term' has to be explicitly specified. Is it with reference to hours, days or months?

Reply: We have clarified that the definition added above applies here as well.

- c) Page 6/62; line- 32-39; The authors mention that ‘This method not only provides us with information about the immediate and longer-term outcome of correct and incorrect predictions, but also overcomes the problems inherent in using the looking while-listening paradigm, as it does not involve pictures of more or less predictable sentence endings, and so the responses cannot be guided by visual input’.

But the primary task in the study is description of animated cartoon characters presented through videos. The participants are instructed to describe it with stem sentence. It does involve visual input and response is guided via visual input after listening to the prime-sentence produced by the investigator. Based on the descriptions given by the authors, we feel the paradigm in the present study is closer to visual (looking) while listening task- there is a need for clarity here.

Clarification may be offered in the report.

Reply: We have added clarification to the report in a footnote on page 5.

- d) ‘There were no error??? verb-based learning effects in general but there was only reliable evidence for immediate prime surprisal effects in the adult, but not in the child group.’ (Page 44/62; line 54-58). Does it suggest that language adequacy probably plays a role in prime-surprisal? As there was no baseline established for language level even in Stage 2 study, the results could have been confounded by this factor.
(We invite input to our understanding if this is incorrect).

Reply: In our opinion the combination of effects mentioned above reflects the limitations of the design used (that our between-participant design was more sensitive to noisier child data, see page 37, and that the study was not designed to test verb-based learning effects, see page 38).

However, our results support a relationship between language adequacy and prime surprisal, as our language baseline measure (TROG) showed an interaction with our prime surprisal measure (interaction of prime structure and verb bias match). However, as this relationship was not the main target of the current investigation, further studies will need to address this question.

- e) Page 25/62: line 17-24: The presence of immediate effects and the absence of lasting effects would show evidence of prediction but no evidence of learning via prediction. Given the complexity of the design and analysis, we suggest authors to supplement such descriptions with examples, wherever possible to avoid wordy narratives in the text.

Reply: We could not modify these lines as they are part of the pre-registered sections.

f) The assumption made by authors with regard to priming, prediction and learning as being aligned on a continuum needs supporting definitions or descriptions. Footnotes or endnotes may be given (if RSOS policy permits). This suggestion is made considering the debate in academia and research regarding the above three constructs. See...

- Dell, G. S., & Ferreira, V. S. (2016). Thirty years of structural priming: An introduction to the special issue. *Journal of Memory and Language*, *91*, 1-4.
- Rowland, C. F., Chang, F., Ambridge, B., Pine, J. M., & Lieven, E. V. (2012). The development of abstract syntax: Evidence from structural priming and the lexical boost. *Cognition*, *125*(1), 49-63.
- Huettig, F., & Mani, N. (2016). Is prediction necessary to understand language? Probably not. *Language, Cognition and Neuroscience*, *31*(1), 19-31.
- Tooley, K. M., & Traxler, M. J. (2018). Implicit learning of structure occurs in parallel with lexically-mediated syntactic priming effects in sentence comprehension. *Journal of memory and language*, *98*, 59-76.
- Jones, T., & Farrell, S. (2018). Does syntax bias serial order reconstruction of verbal short-term memory?. *Journal of Memory and Language*, *100*, 98-122.

Reply: Learning and structural priming are defined on pages 10 and 26 in the manuscript. We have not defined prediction as we are not measuring predictive processes directly (only the learning outcome resulting from these processes). However, as we are targeting the Dual-Path model, our definition of prediction is identical to the one used by Chang, Dell & Bock (2006).

g) Authors claim to test long lasting/long term learning in the study. However, this study did not employ any offline testing to check lasting effect rather the procedure in second post-test was a part of bingo game not much different from the pre-test.

Reply: Our study contrasted immediate priming effects (measured directly after the prime sentence with minimal to no intervening language between prime and target, measured in the test-phase) with lasting effects (that persist over multiple intervening sentences involving those that contain the target structure, here datives, measured in the post-tests). In the current study we did not address how long these learning effects last. Further research will have to establish how long-lasting similar effects are in order to gain a complete picture of the error-based learning mechanism.

h) Syntax learning is not merely a language activity but a cognitive-linguistic activity

(Kurland, 2011). No attempt has been made (at least for children group in which the authors have shown large interest to analyse data) by the authors to examine vigilance and attention, given the fact that the entire task lasted for 45 + minutes in a single session.

Reply: We tested whether children were attending to the task by asking them to repeat all sentences produced by the examiner. If they did not repeat the sentence correctly (even after 3 prompts) the trial was excluded. Participants who had more than half of the trials excluded in the test, post-test and second post-test phases were excluded from the study. Children who did not produce dative sentences in at least half of the trials in these phases were also excluded (see pages 12 and 20). As repeating and producing complex dative sentences is a challenging task for children this age, only including those who carried out this task successfully across all phases of the study ensured that all the participants we included attended to the task.

Minor concerns to be addressed:

1. Page 6/62: Line 22: ‘Visual world paradigm’ – typo error?
To be changed as ‘visual word paradigm’
2. Table 1: Heading
We would like to suggest authors to shorten the heading by keeping the description as Note: below Table or as a small paragraph.
3. Page 14/62; line: 16, Second **pot** test Typo error? To be changed to-Second **post**-test?
4. Page 19/62; line: 31, the rationale for using semi randomised approach may be made clearer (add justification).
5. Page 19; line: 35-40 what is the time gap b/w pre and post and second post-test?
It’s important to note the break or the gap durations.
6. Page 23/62; line: 44- 60? Component ‘c’ is missing?
 - a) bias group
 - b) pre-test
 - c) ???
 - d) age group
7. Sample test items are provided in method. Therefore, the appendix with the elaborate list may be planned as electronic materia1 it is suggested the complete list may be given as electronic.

Reply: 1., 3., 6. are corrected.

2. We cannot change the table heading as it is part of the pre-registered section

4. We have addressed this in footnotes.

5. There was no gap between the study stages.

7. We have removed the sentence lists from the appendix and now they are only included in the supplementary electronic materials.

Appendix F

Fazekas: Do children learn from their mistakes?

The authors are to be congratulated on completing a complex piece of work. The pre-registration, open materials and data, and immaculately presented scripts give confidence in the findings and set a very high bar for other researchers.

I felt the authors did a good job of explaining the logic of the paper and steering the reader through a complex set of analyses and what they meant. The design was clever and the materials very well-constructed. Nevertheless it was not always easy to follow the thread, some analyses seemed overcomplicated, and I had major difficulties the interpretation of results (explained more below).

The methods follow the pre-registration except in some details that are discussed later in the report; the changes all have sensible rationale, namely that the original approach would give ceiling effects.

I appreciate it is not possible to modify the sections that were pre-registered, but I did find it hard to understand why the adult group was included, given that the focus was on how children learn language. Predictions are made that children will show larger effects than adults, but the question really is why would adults show any effects at all, given that they are deemed to be competent language users. Insofar as they do show the predicted effects, doesn't that rather weaken the interpretation of the effect as indicating language learning? This, indeed, relates to my key criticism of the Discussion (see below)

Reply: The current project addressed the notion of error-based learning by testing a specific language acquisition theory, the Dual-Path model (e.g. Chang, Dell & Bock, 2006). This model suggests that the error-based learning mechanism that leads to language acquisition in childhood stays active even after the language acquisition process is complete. In adulthood this same mechanism can help adjusting to different linguistic situations such as different registers or speakers. This is mentioned on page 2 in the Introduction. While children are the main target of our interest, we have included the adult group not only because the theory we tested predicts that adults will show a similar effect to children, but also because adults are easier to recruit and typically provide less noisy data. Thus, by including adults we could gain extra exploratory and statistical power with fewer recruitment and testing issues. The adult data was indeed helpful in understanding the lack of significant immediate prime surprisal effects in the child group.

There are some departures from the specified analysis, but they are well motivated. In particular, the plan had been to measure change from a baseline period in terms of rates of production of sentence frame types, but some participants were already producing 100% at baseline. This makes the use of a change score unsuitable for addressing the research question, and the authors explain this and then present an alternative analysis omitting those who were at ceiling. My preference would be to relegate the original analysis to supplementary materials, since it is not appropriate – the authors can then show it was done, but I think it is important that the paper, already very dense with analyses, does not get bogged down with unnecessary detail. At the end of the day, the point of preregistration is to keep everyone honest, not to make papers indigestible.

Reply: We have relegated the original analyses to supplementary materials and have only kept a short discussion of it in the text as suggested.

Main point of criticism

I found the interpretation of findings in the Discussion was problematic. It is possible that I have misinterpreted some aspect of the design, as it is complex and outside my current research area. But I struggled with the idea that this study is telling us about language learning.

The rationale (line 42) is that 'Every time they make an incorrect prediction, linguistic representations change, which, in children, moves them a step closer to the adult state'. The idea of the study is to assess 'whether less predictable input leads to more lasting change than more predictable input'. Furthermore 'the key prediction of this account is that incorrect predictions lead to learning. To test this, we need to demonstrate that prime surprisal leads to lasting cumulative language change as well.' (p 7)

Yet what was found was that a proportion of both children and adults showed an increase in production of a primed sentence construction when the primed sentence had been presented with a verb that was not usually used with that construction.

The effect was statistically significant but numerically small; assuming I have understood Figure 2 correctly, around 1/3 children showed an effect in the opposite direction to prediction.

Reply: Indeed, not all children showed the expected effect in our study. Thus whether all children use prediction-based learning mechanisms and if so, under what circumstances requires further examination.

A manipulation check (phase 2) failed (section 3.2.3): "neither Hypothesis 1a (replication of prime surprisal effects) nor Hypothesis 1b (larger prime surprisal effects in the child than in the adult group) were supported by the current dataset." Surely this is of concern because it undermines the mechanism that is postulated to explain the gains in phase 3? There is some discussion of this on p. 44.

Reply: The lack of significant immediate prime surprisal effects in the child group is unexpected and requires further investigation. There are multiple possible reasons for the absence of this effect. One possible reason is that the surprisal level was not (sufficiently) different in our sentences, another is that children were not sensitive to surprisal, and a third is that some other characteristic of the design interfered with the prime surprisal effects. These kind of immediate prime surprisal effects have been shown in similar age groups and sentences in another study (Peter et al., 2015) and also in the adult group in our study (in an exploratory analysis). Furthermore, the Bayesian analysis also showed weak evidence for these effects in the child group when baseline DOD was taken into account.

Considering that immediate prime surprisal effects have been consistently detected using stimuli similar to ours, we think it is unlikely that the surprisal-level of the sentences was not different enough or that children were not sensitive to the level of surprisal. In our opinion, it is much more likely that the interaction of the between-participant design and the large variation in baseline level performance was the cause (see pages 36-37).

However, as for the error-based theories the simultaneous detection of a prime surprisal effect and long term-learning is crucial, it is essential that this relationship is examined in further studies.

In a later phase there was a further test where the same verbs were used as had been used

in priming. 'If there is verb-specific error-based learning, we expect an enhanced shift towards the dative structure the verb previously appeared with when the structure did not match the verb's bias.' (p. 10). But in fact, the use of the primed construction dropped away. Thanks to the excellent documentation of scripts and data I was able to look at the data in a slightly different way, and this confirmed that at this 2nd post-test, the effect seen at 1st post-test bounced back to be, if anything, more extreme than the baseline. I may have misunderstood this, but having just looked at means in an initial data check, this did seem inconsistent with expectation.

phase	bgro	agroup	dod mean	sd
	up			
Baseline	D	adult	0.635	0.482
Post-test	D	adult	0.668	0.472
Second post-test	D	adult	0.598	0.491
Baseline	P	adult	0.68	0.467
Post-test	P	adult	0.628	0.484
Second post-test	P	adult	0.74	0.44
Baseline	D	child	0.291	0.455
Post-test	D	child	0.364	0.482
Second post-test	D	child	0.177	0.383
Baseline	P	child	0.355	0.479
Post-test	P	child	0.306	0.462
Second post-test	P	child	0.371	0.484

Reply: The second post-test was not a delayed version of the first post-test, rather the test of verb-based learning (as opposed to the test of abstract learning in the first post-test). While the first post-test featured verbs that were not included in the study before, in the second post-test we repeated the prime verbs from the test phase. As the test-phase had a between-participant design (group A only heard DOD-biased verbs, and group B PD-biased verbs), participants consistently produced sentences with different verbs in the second post-test too. This meant that participants in group A only produced sentences with DOD-biased verbs in the second post-test, while participants in group B always used PD-biased verbs. As the verb-bias of the target-verb influences the likelihood of DOD-production (see e.g. Peter et al., 2016) the DOD-production difference in this phase likely reflects this target-verb-bias difference. This also means that in the second post-test the DOD production is not directly comparable in the two bias groups. (Note that in the first post-test the verbs and thus the verb-bias level of the target verbs was identical, thus this problem is not relevant there and the DOD level can be directly compared in the two bias groups.)

A related question is whether the temporary shift in preference for one construction over another at first post-test can be regarded as 'moving a step closer to the adult state'. It's unclear how this could be given that we are already told that 'Children of this age consistently produce both PD and DOD structures (with an average DOD production of approximately 30%) in corpus-based studies.' So in what sense have they not already attained 'the adult state'. And the adults are presumably already in 'the adult state'.

Reply: The study aims to model the learning mechanism by shifting the participants towards an artificial distribution (more DODs or more PDs depending on bias group)

rather than moving participants towards the adult state. We claim that the same mechanism that resulted in participants moving closer to the artificial distributions we set them, is also the one that moves children closer to the adult state and adults closer to any novel language environment that they encounter.

So it is not clear that these results are relevant for understanding language *learning*. What has been shown is that priming, and predictability of structural variants, can induce a temporary shift in preference for one version of a construction than the other, in children who already are familiar with both forms.

The study can be regarded as demonstrating, therefore, that children have some sensitivity to the 'surprisingness' of incoming sentences, but I do not think that this has been shown to influence language learning. It looks more like a kind of delayed priming of preference for one form over another – and given the results from the second post-test, this does seem temporary. This result is itself of interest.

Reply: Within the Dual-path model (that is tested here), any non-immediate priming effect (even one that lasts for only a few intervening sentences) must reflect long-term weight changes (i.e. learning), since immediate activation effects decay instantly. In other words, learning is defined as weight change in the model and delayed priming is a manifestation of that weight change.

We hope that we have addressed the individual comments and questions about the study's relation to learning above.

More minor points

I find the title non-optimal. I had assumed from the title that there would be some analysis of children's mastery of grammar in relation to their making grammatical errors. This kind of thing is discussed in relation to prior literature on p 3. But this study is not about children making linguistic errors: rather, it's about them learning from hearing familiar constructions in surprising sentence contexts. The 'error' that is referred to is assumed: a mismatch between prediction and what is heard. This is pretty confusing. So I think a more accurate title would be 'Do children learn more from unpredictable input?' or 'Do children learn more when input does not match predictions?'. Though given my reservations about what has been shown, maybe avoid mentioning 'learning' too, e.g. – 'Are children sensitive to predictability of language input?'

Reply: To avoid misunderstandings about linguistic mistakes we have changed the title to "prediction mistakes" however we have kept "learn" in the title as the study is testing a theory of language acquisition.

One change from pre-registration is that The Test for Reception of Grammar was substituted as a covariate to assess language ability instead of the Sentence Imitation task, because the latter gave ceiling effects. That is adequate justification. However, I would ask that one clarification be made that would involve modifying the original pre-registration. It is unclear which version of TROG was used (the test is not referenced; is this TROG or TROG-2) and how it was scored. The usual scoring is in number of blocks correct, with a range from 0 to 20. This gives a raw score which increases with age between 5 and 6 years. This can be converted in to an age-scaled score or percentile. We need to know which of these was used, especially as the presented data appear to be on a different scale. The Sentence Imitation task also should be referenced.

Reply: We have added the requested information to the manuscript on pages 18-19.

7. Table 7: it would help if the labels for the phases were added, to match Table 1.

Reply: We have added the labels to the phases in Table 7 (now Table 6.).